# Layer 6 ensembles can selectively regulate the behavioral impact and layer-specific representation of sensory deviants

Jakob Voigts[1,2]*, Christopher A Deister[1], Christopher I Moore[1]*

[1]Department of Neuroscience and Carney Institute for Brain Science, Brown University, Providence, United States; [2]Department of Brain and Cognitive Sciences, MIT, Cambridge, United States

**Abstract** Predictive models can enhance the salience of unanticipated input. Here, we tested a key potential node in neocortical model formation in this process, layer (L) 6, using behavioral, electrophysiological and imaging methods in mouse primary somatosensory neocortex. We found that deviant stimuli enhanced tactile detection and were encoded in L2/3 neural tuning. To test the contribution of L6, we applied weak optogenetic drive that changed which L6 neurons were sensory responsive, without affecting overall firing rates in L6 or L2/3. This stimulation selectively suppressed behavioral sensitivity to deviant stimuli, without impacting baseline performance. This stimulation also eliminated deviance encoding in L2/3 but did not impair basic stimulus responses across layers. In contrast, stronger L6 drive inhibited firing and suppressed overall sensory function. These findings indicate that, despite their sparse activity, specific ensembles of stimulus-driven L6 neurons are required to form neocortical predictions, and to realize their behavioral benefit.

**\*For correspondence:**
jvoigts@mit.edu (JV);
Christopher_Moore@brown.edu
(CIM)

**Competing interests:** The authors declare that no competing interests exist.

## Introduction

The six-layered architecture of mammalian neocortex emerged relatively late in evolution. A commonly proposed function of neocortex is to form dynamic predictive models that include expectations of incoming stimuli and contextually appropriate actions. The ability to generate and update these models of the world enables rapid and adaptive shifts in behavior that underlie cognitive capabilities such as flexible language.

An elemental example of model formation is evident in the response to deviations from ongoing patterns. When identical sensory stimuli are repeated and then a 'deviant' occurs, neocortical neurons often fire differently than they would to the deviant in isolation, or after its repetition (*Chater et al., 2006*; *Rao and Ballard, 1999*). Signatures of this computation are found in visual (*Courchesne et al., 1975*), auditory (*Tiitinen et al., 1994*; *Ulanovsky et al., 2003*), and language-processing (*Kutas and Hillyard, 1980*) areas. Such change detection is typically studied as the increase in firing rates elicited by deviant stimuli following stimulus-specific adaptation (*Movshon and Lennie, 1979*). Stimulus-tuned neurons along the afferent pathway, including at thalamocortical synapses, adapt to repeated stimulation (*Chung et al., 2002*; *Katz et al., 2006*; *Khatri and Simons, 2007*) and subsequent deviant stimuli activate new pools of less adapted neurons, leading to increased neocortical drive.

However, in addition to such bottom-up adaptation mechanisms, neocortically-represented factors such as stimulus context, history, and expectation (*Knierim and van Essen, 1992*; *Chelazzi et al., 1993*; *Reynolds et al., 2000*; *Maunsell and Treue, 2006*) also influence sensory responses, supporting the hypothesis that neocortical representations could be key to deviant

processing. While likely all neocortical layers contribute to model implementation, layers 2/3 (L2/3) are a leading candidate for the representation of more complex information. These supragranular layers are typically the first to express receptive-field plasticity in response to sensory change, prior to L4 (*Diamond et al., 1994*), and are more susceptible to modulation by shifts in attentional state than deeper layers (*Hyvärinen et al., 1980*). Further, in primary somatosensory and visual neocortices, L2/3 receptive fields can encode specific temporal stimulus patterns (*Estebanez et al., 2016*; *Hubel and Wiesel, 1968*), and can integrate multiple types of information (e.g. motor and sensory signals) and the mismatch in their alignment (*Zmarz and Keller, 2016*).

Layer 6 (L6) is also well positioned to contribute to the neocortical implementation of predictive models, as it integrates lemniscal thalamic, long-range cortico-cortical, and modulatory inputs (*Martin and Whitteridge, 1984*; *Zhang and Deschênes, 1998*; *Thomson, 2010*; *Zhang et al., 2014*; *Vélez-Fort et al., 2014*). Corticothalamic L6 neurons (CT) in primary visual and somatosensory neocortex are sparsely sensory driven (*Lee et al., 2008*) with selective receptive fields (*Vélez-Fort et al., 2014*) and can robustly modulate sensory gain through an intracortical pathway (*Olsen et al., 2012*). These findings suggest that specific populations of L6 neurons could regulate sensory responses depending on stimulus context. This prediction is supported by L6-mediated modulation of visual receptive fields by stimulus context (*Bolz and Gilbert, 1986*), by the relay of information from head-motion into V1 via L6 (*Vélez-Fort et al., 2018*), and by preferential involvement of deep cortical layers in top-down sensory processing (*Kok et al., 2016*). However, whether L6 contributes to the representation of stimulus changes across layers, and to perception, is not known. Here, we tested this hypothesis using selective optogenetic modulation of L6 activity in awake behaving mice, single-neuron recordings across neocortical layers, and 2-photon Ca$^{2+}$ imaging of genetically identified L6 cells.

## Results

### Weak drive of L6 CT negates the behavioral impact of deviants, but does not affect baseline sensory performance

We tested the impact of manipulating L6 CT cells in a naturalistic and untrained, but well studied and characterized sensory decision-making task, Gap Crossing (*Hutson and Masterton, 1986*; *Figure 1a*). In this task, mice use their vibrissae to locate and cross between elevated platforms whose distance is changed after each trial (~4–6 cm, six mice, *Figure 1—figure supplement 1b*). Experiments were performed under near infrared illumination and auditory white noise to reduce

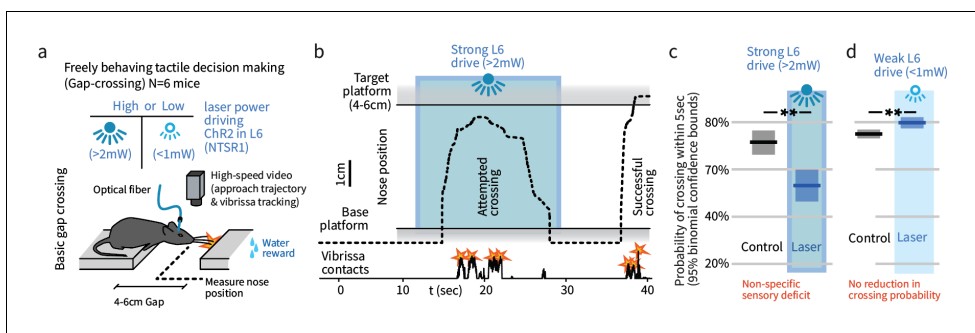

**Figure 1.** In a natural behavior, strong optogenetic drive of L6 causes a sensory deficit. (**a**) To test the impact of L6 modulation in a naturalistic context, we used an unrestrained Gap Crossing task and applied high or low power optogenetic drive to L6 CT cells (ChR2 in NTSR-1 cre line). (**b**) Example trace (nose position over the gap) where strong optogenetic L6 drive disrupted Gap Crossing behavior, causing the mouse to retreat before crossing. (**c**) Strong L6 drive made mice less likely to cross the gap (N = 514 trials, bootstrapped 95% confidence intervals (CIs)). (**d**) In contrast, weaker L6 drive did not reduce crossing probability (N = 751 trials) but led to a small increase in crossing probability.

The online version of this article includes the following figure supplement(s) for figure 1:

**Figure supplement 1.** Overview of trial structure and timing for the experimental paradigms used in this study.

visual or auditory confounds (*Hutson and Masterton, 1986*; *Voigts et al., 2008*). Only trials with crossings within 5 s of exploring the gap were analyzed. We applied selective optogenetic depolarization to L6 in mice expressing Channelrhodopsin (ChR2) in L6 corticothalamic (CT) pyramidal cells (GN 220-NTSR1 Cre line [*Gong et al., 2007*]).

We tested mice on the Gap Crossing task under two conditions, strong L6 CT optogenetic activation (>2 mW power) that recruits trans-laminar inhibition and reduces neocortical sensory gain in V1 (*Olsen et al., 2012*), or weak L6 drive (<1 mW) (*Figure 1a*). Strong L6 drive led to a non-specific sensory deficit, reducing the likelihood of successful Gap Crossing from ~75% to ~50% (p<0.01, *Figure 1b,c*), but had no detectable effect on the whisking pattern of the mice (*Figure 2—figure supplement 1*).

To test the effect of L6 drive on spontaneous stimulus change, we induced small deviations during active palpation (*Voigts et al., 2015*) (the 'Gap Crossing with Deviation' task). Mice were not rewarded for detecting this deviation. In ~35% of trials, the target platform was rapidly pulled back by ~2 mm during a bout of exploration (*Voigts et al., 2015*; *Figure 2a*). To control for non-specific cues that could have been associated with this retraction (e.g. sound or air currents), true 'change' trials in which the vibrissae contacted the retracting platform both before and after the retraction (as illustrated in *Figure 2b*, black, N = 317) were compared to 'sham-change' trials in which they contacted the platform either only before (light green, N = 49) or only after (dark green, N = 134) a platform motion. On sham trials, mice could not perceive a change in position through vibrissal palpation but would experience any other effects of platform motion. In the true change condition, but in neither of the sham conditions, mice slowed their approach (*Figure 2c*, p<0.005 change vs. only before, and p<0.01 change vs. only after, rank sum), presumably to precisely re-locate the target before crossing, showing that mice perceive and react to the sensory deviant.

We next tested whether the weaker L6 optogenetic drive (<1 mW) would impair this change detection behavior, and found that it removed the extra sampling time that platform motion would otherwise generate (p<0.001 laser vs. control, N = 317 trials, *Figure 2d,e*, *Video 1*). Weak L6 drive did not affect behavior when the target platform was static (p=0.9, N = 751 trials, *Figure 2f,g*, *Figure 2—figure supplement 1*). The weaker drive also slightly increased the overall likelihood of crossing (*Figure 1d*, same analysis as *Figure 1c*). Gap Crossing is sensitive to changes in sensory function (*Celikel and Sakmann, 2007*; *Chaudhary and Rema, 2018*), the maintenance of regular Gap Crossing behavior in the static platform condition indicates that weak L6 CT drive did not impact basic sensory function, but did impact change detection.

Detecting a sudden change in platform position in the Gap Crossing with Deviation experiment likely results from a combination of multiple sensory parameters that change because of the changed position of the vibrissa contacts. Previous high-speed videography studies under the same experimental conditions showed that these parameters include a change in amplitude of vibrissal contact deflections (*Voigts et al., 2015*), as well as other parameters such as relative timing (*Voigts et al., 2015*), and likely deflection angles (*Knutsen et al., 2008*; *Ritt et al., 2008*), assymetry (*Dominiak et al., 2019*), time course of the torque (*Pammer et al., 2013*), and vibrations (*Ritt et al., 2008*; *Lottem and Azouz, 2009*).

To test whether the impact of weak L6 drive on the effects of sensory deviation replicates in a head-fixed behavior in which one specific parameter of vibrissal motion is changed, mice were trained to detect vibrissal motion (*Figure 3—figure supplement 1*), applied as a series of deflections (*Stüttgen and Schwarz, 2008*; *Sachidhanandam et al., 2013*; *Miyashita and Feldman, 2013*; *Siegle et al., 2014*). A subset of these trains contained direction deviants (*Figure 3a*), though mice were not trained to report the presence of a deviant. Deviants increased the detection rate for threshold level stimuli (*Figure 3a,b*). Weak L6 drive removed this effect but had no impact on detection of stimuli containing no deviant. As in Gap Crossing with Deviation, altering L6 activity through weak optogenetic drive selectively disrupted the behavioral benefit of these stimulus changes. We also observed that weak L6 drive disrupted encoding of deflection direction in L6 and of deviant stimuli in L2/3 (*Figure 3—figure supplement 2*).

## Weak drive of L6 CT changed the identity of stimulus-driven ensembles in L6 without changing mean firing rates across layers

We next quantified the impact of strong (disrupts Gap Crossing) and weak (selectively disrupts the impact of deviants) optogenetic L6 drive regimes on basic sensory encoding in head-fixed mice. To

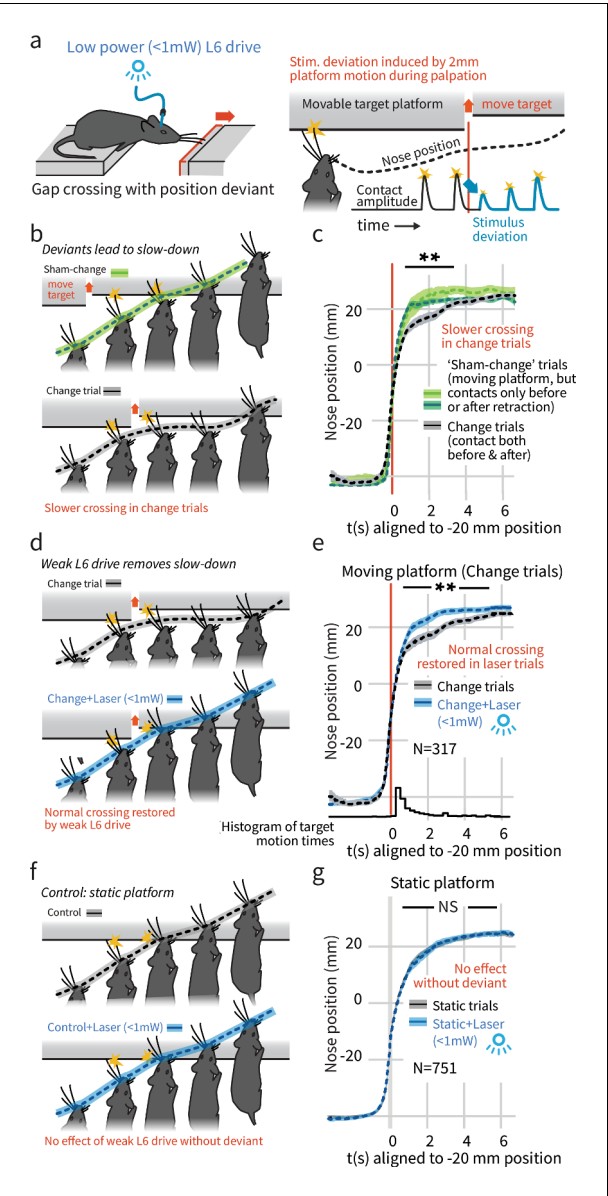

**Figure 2.** In a natural behavior, weak L6 drive selectively negates processing of deviant stimuli. (a) To create a sudden, small sensory deviation, in the Gap Crossing with Deviation task the target platform was retracted by ~2 mm during bouts of tactile sampling. Prior high-speed videography studies have shown this small, unexpected deviation in the position of the platform leads to changes in vibrissal motion, including reduced deflection amplitudes (Figure 9 of *Voigts et al., 2015*). (b) When the platform was retracted in the middle of a bout of vibrissal contacts (*black trace*), mice slowed their approach relative to control trials where they contacted the target either exclusively before (*light green*) or after (*dark green*) platform retraction. (c) Summary statistics (95% CIs) of mouse position over time in the three trial types. (d) Weak L6 drive removed the deceleration associated with platform position deviants. (e) During weak L6 drive, the time course of the Gap Crossing with Deviation task does not show the signature deceleration created by a change in platform position during palpation. (f, g) Weak L6 drive had no effect on the Gap Crossing when the platform was static, showing that it selectively abolished the reaction to platform position deviants with little effect on other sensory and sensorimotor function. See *Figure 2—figure supplement 2* for per-mouse data.

The online version of this article includes the following source data and figure supplement(s) for figure 2:

**Source data 1.** Gap crossing speed across sham-change, control, and L6 drive conditions.
**Figure supplement 1.** Per-mouse analysis of freely behaving gap-crossing and sham-laser control experiment.
**Figure supplement 2.** Whisking pattern kinematics in the gap-crossing task are not impacted by laser stimulation.
**Figure supplement 2—source data 1.** Whisking dynamics for the Gap rossing task.

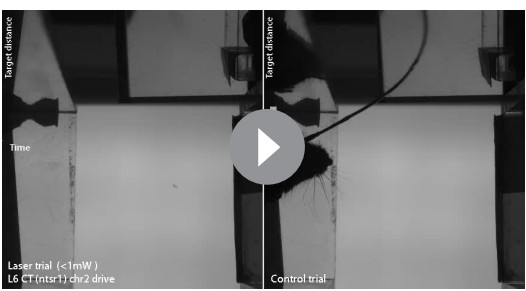

**Video 1.** Mouse crossing in the Gap Crossing with deviation task. Example raw videos from two trials. Left: Weak L6 drive leads to mice not reacting to sudden small platform position deviations. Right: Control trial, mice slow down their approach in reaction to sudden platform position deviations. Inserts show target distance over time.

https://elifesciences.org/articles/48957#video1

approximate the stimulus statistics that occur during Gap Crossing in head-fixed mice, we delivered trains of vibrissa deflections (seven deflections, 10 Hz) to the B and C row vibrissae, arcs 1–3, and recorded in matched somatotopic positions in vibrissal primary somatosensory neo-cortex (SI). Vibrissae were clamped in the piezo-stimulator (see Materials and methods) and mice did not whisk. Prior studies have shown that strong L6 drive reduces sensory gain (*Olsen et al., 2012*; *Denman and Contreras, 2015*) in V1, and we observed analogous suppression in SI when we drove L6 with the strong optogenetic stimulus (*Figure 4a*, acute laminar probe recording).

We next investigated sensory responses under weak L6 drive using chronic tetrode recordings in awake mice. To characterize layer-specific activity, we implanted high-density arrays of movable multi-contact electrodes (*Voigts et al., 2013*), stereotrodes and tetrodes, in SI, yielding ~25 identified neurons per session (five mice, *Figure 4—figure supplement 1*). We recorded 1242 neurons over 75 sessions, 395 were phasically stimulus-driven (see Materials and methods). To classify neurons by layer, we tracked electrode depth and stimulus-evoked LFP (*Castro-Alamancos and Connors, 1996*), and categorized their spike waveform (*Bortone et al., 2014*) as regular spiking ('RS'; typically excitatory pyramidal neurons) or fast-spiking ('FS'; typically inhibitory interneurons: ~30% were FS). Consistent with prior studies (*Thomson, 2010*; *Lee et al., 2008*; *Swadlow and Hicks, 1996*), L6 neurons were sparsely sensory responsive, as only 18 (RS and FS) of 139 (~13%) were phasically sensory driven. Further, driven L6 neurons were also sparse in their response rates, showing only small vibrissa-driven increases in their activity (*Figure 4b*).

We applied weak L6 drive as during Gap Crossing (~0.1–0.5 mW total) and ramped the light intensity over >100 msec prior to sensory stimulation, to prevent overlap of sensory responses and transient changes in firing rates during initial light exposure. In contrast to stronger drive, this stimulus did not impact sensory-evoked firing rates in RS from any layer (*Figure 4b*, *Figure 4—figure supplement 2*). The only significant impact on sensory responses was a modest decrease in sensory responses in L4 FS (31.7 vs. 24.6 Hz median peak rates, p=0.001, *Figure 4b*). A small increase in firing rates in non-sensory-responsive L6 FS was also observed (*Figure 4b*, p=0.02 sign-rank, laser - control firing rates, N = 37). Increased activity in these L6 FS presumably offset the direct effects of optogenetic drive in L6 RS, keeping L6 RS firing rates at a baseline level (p=0.18, N = 89, signed rank). In L2/3 and L5 RS, transient suppression was evident after optogenetic stimulus onset, but rates returned to baseline prior to sensory stimulation. We did not observe slower changes in the effect of this L6 drive over the time course of the stimulus train.

Extracellular tetrode recordings in L6 do not yield the sample sizes required to characterize their sensory responses (*Figure 4b*), and cannot distinguish between corticothalamic (CT) pyramidal cells hypothesized to contribute to change detection, and corticocortical (CC) cells that are less specifically tuned and lack long-range input from higher cortical areas (*Vélez-Fort et al., 2014*). We therefore employed 2-photon calcium imaging of genetically identified L6 CT cells in GN 220-NTSR1 Cre mice (*Gong et al., 2007*) expressing GCaMP6s (*Chen et al., 2013a*; *Figure 5*, N = 3408 cell bodies imaged; N = 2685 tested with amplitude deviants, in six mice). To study the effect of weak optogenetic drive on L6 population activity, we combined wide-field optogenetic stimulation with 2-photon imaging (N = 1183 cells in four mice). Initial attempts to image L6 somata in mice with transgenic GCaMP6s expression, obtained by crossing the NTSR1 Cre and GCaMP6s reporter lines, had low signal-to-noise due to the dense fluorophore expression in more superficial processes (*Theer et al., 2003*). To obtain suitable image quality (*Figure 5a–c*), sparse labeling of L6 CT neurons was achieved by viral transduction (see Materials and methods). Using this approach, we were able to

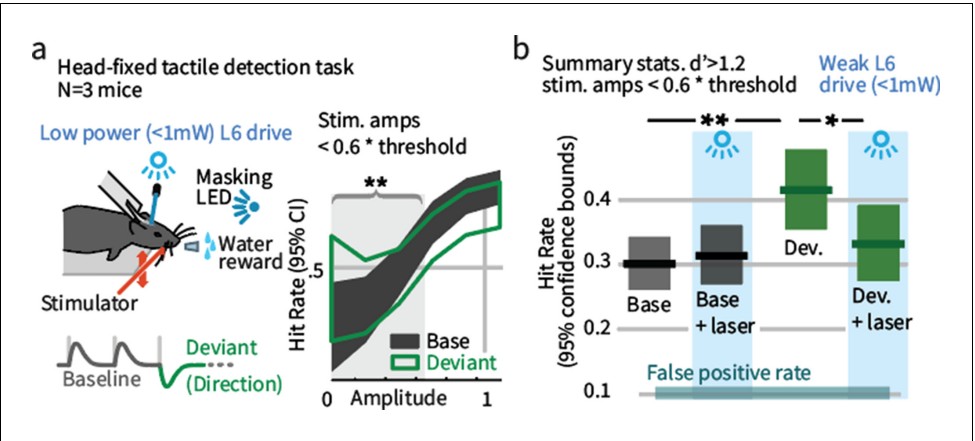

**Figure 3.** Weak L6 drive negates the improved sensitivity driven by inclusion of deviant stimuli in a head-fixed detection task. (**a**) Left: Mice were trained on a go/no-go stimulus detection task. Mice were not explicitly trained to report the presence of a deviant. In 50% of trials, direction deviants were present at positions 2–4 in the stimulus train. Right: Example psychometric curve from one mouse. Deviants increased detection rates for weak stimuli that were less than 60% of the threshold amplitude (*gray region*). (**b**) Box plots show 95% CIs (via bootstrap) for the detection probabilities for this range of stimuli from three mice for baseline and deviant-containing stimuli, for the control (blue masking light) and laser (weak L6 drive, <1 mW) condition.

The online version of this article includes the following figure supplement(s) for figure 3:

**Figure supplement 1.** Head-fixed behavior depends on vibrissa stimulation and is independent of other (auditory or visual) cues.

**Figure supplement 2.** Encoding of directional deviants, and effect of L6 activation.

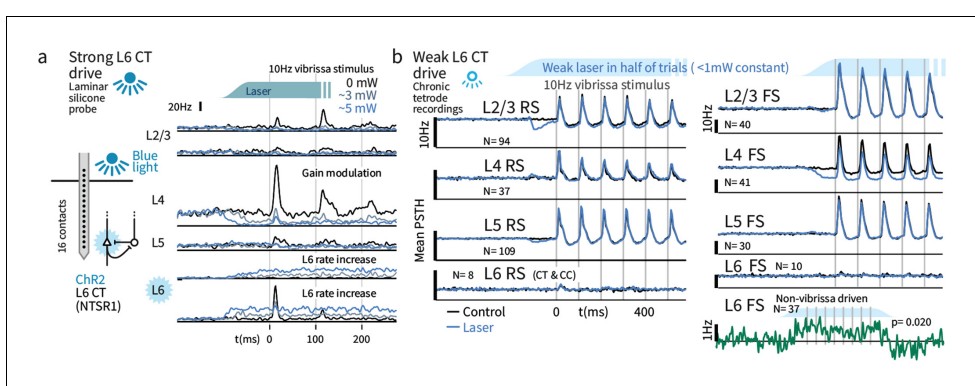

**Figure 4.** Weak optogenetic stimulation of L6 CT has almost no detectable impact on firing rates across layers and cell types. (**a**) Strong optogenetic L6 CT drive increased L6 firing rates and reduced sensory gain in other layers, as in visual neocortex (*Olsen et al., 2012*; *Bortone et al., 2014*), measured with laminar probes. Circuit diagram adapted from *Olsen et al., 2012*; *Helmstaedter et al., 2008*. (**b**) Chronic tetrode recordings across cortical layers. Low power (<1 mW) optogenetic drive did not change mean sensory evoked firing rate of RS in any layer, though a prestimulus change was observed in L2/3. Similarly, FS rates remained largely intact, with the exceptions that small firing rate changes were observed in L4 FS (decreased activity) and L6 FS that were not sensory responsive (increased activity).

The online version of this article includes the following source data and figure supplement(s) for figure 4:

**Source data 1.** Extracellular electrophysiology responses.
**Source data 2.** Extracellular electrophysiology spike data.
**Figure supplement 1.** Proportion of fast spiking (FS) and regular spiking (RS) neurons across cortical layers.
**Figure supplement 2.** Peak firing rates (peak responses for deflections 2-7 for each neuron) for control and laser conditions across all stimuli.

image throughout the upper ~100 μm of L6 (*Figure 5b,c*, *Figure 5—figure supplement 1*), and signal-to-noise ratios exceeded 150% ΔF/F (*Figure 5d*).

As in the electrophysiological data, stimulus-evoked L6 activity was sparse (*Figure 5—figure supplement 2*), and overall L6 calcium signals (relative firing rates) remained unchanged during weak optogenetic drive (*Figure 5f*, p=0.10, signed rank). However, weak optogenetic manipulation changed L6 receptive fields: Individual L6 cells showed suppression or facilitation of vibrissa-evoked responses, including both emergence of newly stimulus-driven cells and the complete removal of sensory responses (*Figure 5g,h*). As such, while the net spiking output of L6 did not change, the specific ensemble of sensory-responsive cells changed substantially with this manipulation. As such, this level of optogenetic drive provides a direct approach for perturbing L6 sensory ensembles without detectably altering L6 firing rates. Due to the slow timescale of the GCaMP6s indicator, we did not analyze responses to individual deflections.

## Small variations in baseline stimulus amplitude did not change firing rates in layers 4 and 2/3 of SI

To understand how shuffling the active ensemble of L6 CT cells selectively affects the processing of small stimulus deviations (*Figures 2* and *3*), we asked how vibrissa stimuli are encoded throughout

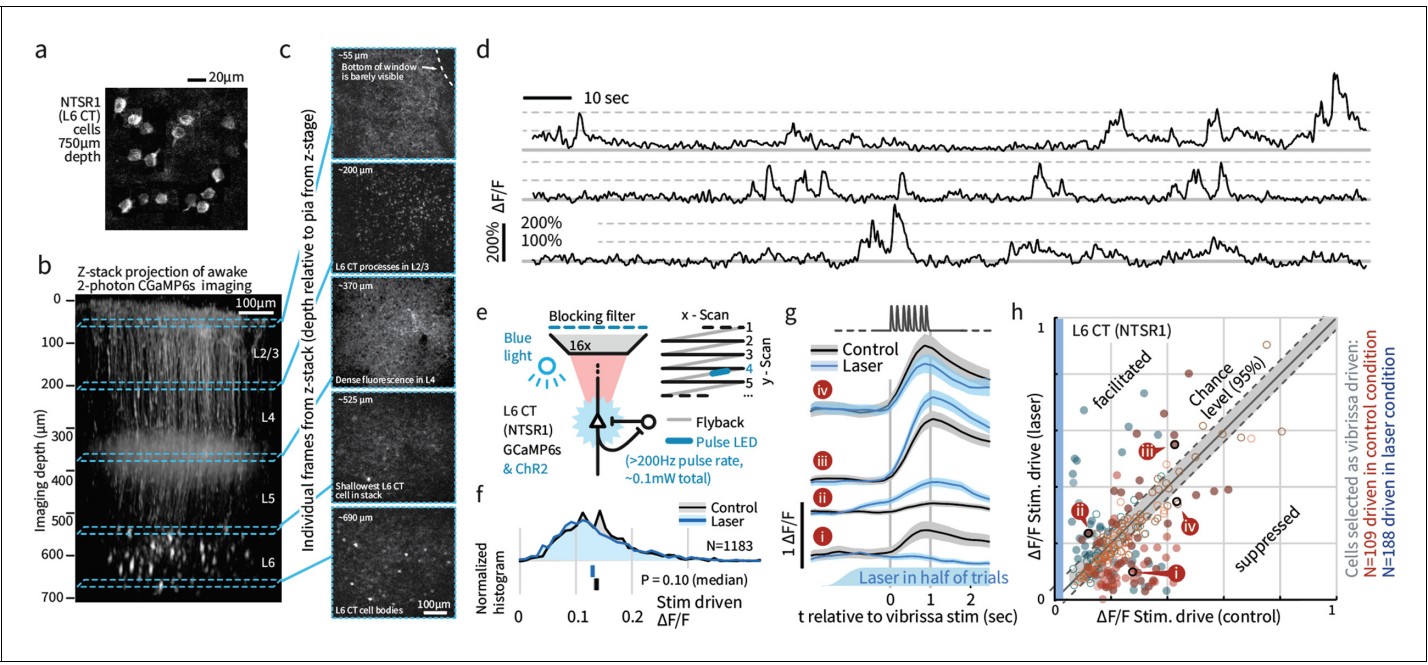

**Figure 5.** Weak depolarization of L6 CT maintains overall rates but changes the identity of stimulus-driven ensembles in L6. To test the impact of weak optogenetic drive on L6 CT, we combined optogenetic stimulation with awake 2-photon imaging using GCaMP6s expression in the L6-specific NTSR1-Cre line (see Materials and methods). (a) Example image of L6 CT somata, image summed over 12 individual frames, in each frame a subset of the cells were active. (b) An example Z-stack projection of GCaMP6s expression in L6 CT. (c) Sample frames from different depths of the Z-stack. (d) Example time series showing typical signal-to-noise ratios. (e) Optogenetic activation was interleaved with imaging at >200 Hz during laser scanning 'flyback'. (f) Optogenetic drive did not significantly change the mean population response of L6 to sensory input, ranksum test, CI via bootstrap of median. (g) However, individual L6 cells showed significant facilitation or suppression of vibrissa-evoked responses during optogenetic drive. (h) Optogenetic L6 activation did not change the overall output from L6, but shifted the identity of the activated ensemble (p<0.001 IQR vs. shuffled control), regardless of whether cells were classified as vibrissa-driven in the control (*red*) or in the laser (*blue*) condition. Filled circles indicate cells where the laser effect was individually significant per-cell (p<0.0025, at 1 s time point, via bootstrap). Of cells that were sensory responsive in the laser condition, 25.0% were significantly facilitated, 23.9% suppressed, and of cells sensory responsive in the control condition, 10.9% were facilitated, 32.9% suppressed.

The online version of this article includes the following source data and figure supplement(s) for figure 5:

**Source data 1.** L6 effect of weak optogenetic drive.

**Figure supplement 1.** Histology and 2-photon imaging in L6 CT cells with GCaMP6s in the NTSR1-Cre line.

**Figure supplement 2.** Classification of L6 CT cells as vibrissa-stimulus driven.

**Figure supplement 2—source data 1.** L6 calcium imaging sensory response statistics.

the cortical column. We delivered trains of vibrissa deflections (seven deflections, 10 Hz) to the B and C row vibrissae of awake head-fixed mice. To approximate the stimulus deviations in the Gap Crossing with Deviant experiments, where mice determine distances to objects via small differences in the amplitude and velocity, and likely other parameters, of vibrissa contacts (*Voigts et al., 2008*; *Voigts et al., 2015*), we selected a narrow range of randomized baseline stimulus amplitudes, and inserted deviants in stimulus amplitude, ±5–15% of baseline (baseline ~ = 25 mm/s, N = 72 sessions). We chose amplitude deviants because they are a significant factor in the Gap Crossing with Deviants experiment (*Voigts et al., 2015*), provide a single parameter that can be kept close to neutral, and keeps the timing of the stimuli identical. This stimulus design thus has two parameters, baseline amplitude and deviant amplitude, with deviants varying as an increase or decrease.

We analyzed the firing probability per vibrissa deflection in our chronic extracellular recordings, as a function of baseline amplitude and deviant amplitude. Variations in stimulus amplitude (without deviants) were not encoded in the mean firing rates of layer 4 (L4) or L2/3 RS cells (*L4* N = 37 RS; spike count difference between larger and smaller stimuli, signed rank p=0.221, 95% confidence interval (CI): [−0.22, 0, 0.002], *L2/3* N = 94, p=0.201, CI:[−0.009,–0.003,0], *Figure 6a–c*). Small variations in overall stimulus strength were therefore not reliably encoded in SI.

## Presence of amplitude deviants was not typically reflected by mean rate increases

We next asked if deviations in stimulus amplitude following repeated baseline deflections at a fixed amplitude (analogous to the sensory deviants induced by the small change in object distance in the Gap Crossing with Deviation task) were reflected in firing rates. The most commonly reported effect of stimulus deviants is an increased neocortical response (*Movshon and Lennie, 1979*; *Khatri and Simons, 2007*) regardless of deviant identity. In this case we would observe neurons that increase their rates when stimuli decrease or increase in amplitude. To test for such encoding, we grouped all deviants (increases and decreases) and examined their grouped mean effect on firing rates. While individual neurons displayed such sensitivity to deviation (*Figure 6d*), neither L2/3 nor L4 neurons systematically increased their overall firing rates for deviants when amplitude increases and decreases were grouped together (*Figure 6e*), showing that our stimulus design avoided pre-cortical stimulus-specific adaptation (*Movshon and Lennie, 1979*).

## Layer 4 neurons encoded stimulus deviations with positive 'change coefficients'

While overall firing rate did not change when summed across deviants (both increases and decreases), we further explored how deviation type impacted firing (increase vs. decrease). A neuron might for instance increase its rate for deviants where stimulus amplitude increases and decrease its rate for amplitude decreases. *Figure 7a* shows examples of the types of responses found, including significantly greater firing in response to deviants of increasing amplitude (examples i and iv), and the opposite, increased responses to deviants that decreased in amplitude (ii and iii). To quantify these effects, we calculated the spike count difference between responses to deviants with increased or decreased amplitude, per cell, per deflection. We term this difference a 'change coefficient,' with greater firing to amplitude increases referred to as a 'positive' change coefficient, and vice versa. Only the first deviant stimulus in each train was analyzed, and baseline control deflections were chosen from corresponding positions in non-deviant-containing stimulus trains.

L4 neurons consistently showed positive change coefficients, firing more spikes for deviant increases and fewer for decreases. This encoding is evident at the population level as a rightward shift in the distribution of change coefficients (*Figure 7b*, p=0.005, CI:[0.012, 0.016, 0.035]; See *Figure 9—figure supplement 1* for generalized linear model 'GLM' analysis). These spike count differences for deviations were larger than for equivalent variations in baseline amplitude with no deviant (p=0.016, signed rank). We observed no significant deviant encoding in L5 (N = 92: p>0.05). These findings in L4 can be interpreted as enhanced specificity of encoding following an adapting stimulus, as observed across several stimulus types (*Ollerenshaw et al., 2014*; *Moore et al., 1999*; *Andermann et al., 2004*), in this case amplitude encoding.

While these effects were small when considered for single neurons and single, small amplitude vibrissal deflections (<0.1 spikes/deflection), they are substantial when considered across

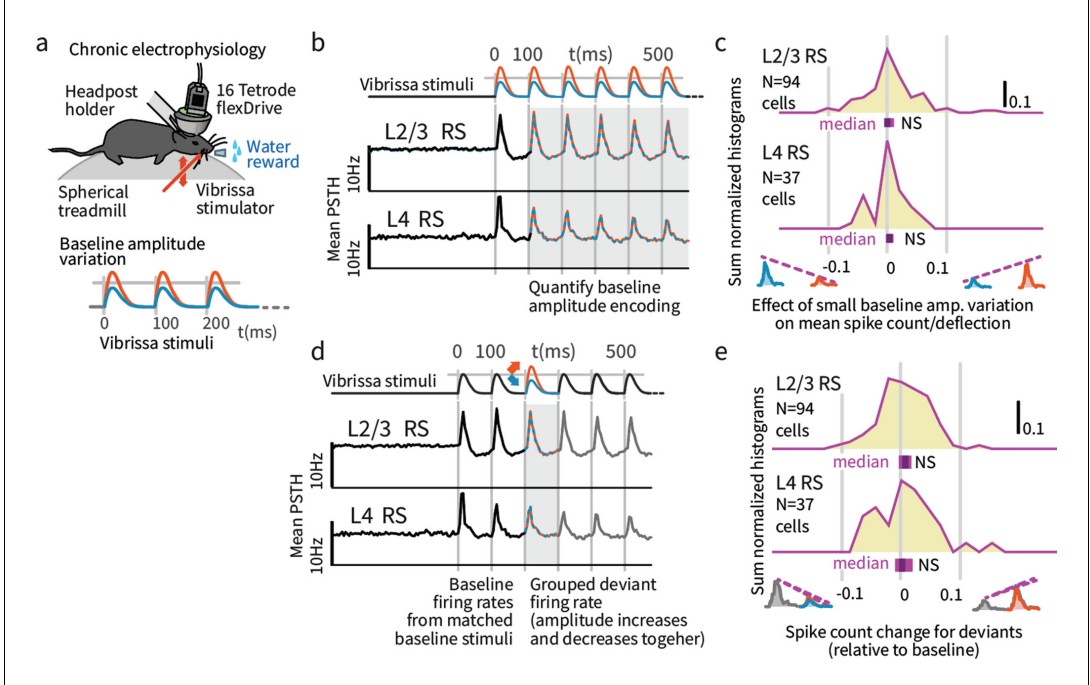

**Figure 6.** Small variations in baseline stimulus amplitude were not robustly encoded in layers 4 and 2/3 SI. (a) Experimental preparation. Baseline stimulus amplitudes were varied by up to 15% on a trial-by-trial basis, in a range of ± ~ 25 mm/second, and presented in trains of 7 stimuli at 10 Hz. (b) The average sensory-evoked PSTHs from sensory-responsive RS neurons showed that neither L2/3 nor L4 cells significantly changed their mean firing rates to reflect stimulus amplitude in the range employed. (c) The population distribution of differences in spike rates per deflection between small and large baseline stimulus amplitudes showed no significant encoding of amplitude. *Purple bars* beneath each histogram indicate 95% CI (via bootstrap of median), ranksum test. (d) To probe the encoding of deviant presence, we presented amplitude deviations in the middle of stimulus trains. (e) To test for encoding of deviant presence, regardless of whether the deviants were increases or decreases in deflection amplitude, we calculated the average effect of deviants on firing (both types grouped, increases and decreases). While a few individual cells showed a generalized sensitivity to deviation, amplitude deviants did not systematically affect overall firing rates in L2/3 or L4, as shown in the centered population distribution.

The online version of this article includes the following source data for figure 6:

**Source data 1.** Spike data on effect of weak L6 drive on stimulus change representation.

populations of neurons, and in the context of vibrissal deflections during natural exploration and Gap Crossing. This increased sensitivity of amplitude encoding in L4 after adaptation to a baseline is consistent with the enhanced discriminability of stimulus features generally observed with adaptation (*Ollerenshaw et al., 2014*; *Békésy, 1967*; *Tannan et al., 2006*; *Abbott et al., 1997*).

## Layers 2/3 neurons explicitly encoded stimulus amplitude deviations

We next examined change encoding in L2/3 neurons (N = 94 vibrissa-driven neurons out of 363 recorded). As in L4, a subset of L2/3 neurons showed positive change encoding, but negative change coefficients were also observed, i.e. neurons with greater firing in response to stimulus decreases (*Figure 7a*, examples ii and iii). At the population level, significant heterogeneity would be reflected in a broadening of the change coefficient distribution relative to non-deviant stimuli. To test whether this heterogeneity was significant at the population level, we computed a surrogate distribution from shuffled baseline stimuli (*Figure 7b*, gray). The observed distribution was significantly broader than the surrogate (interquartile range/IQR; *Figure 7c*; p=0.006, Shannon entropy: p=0.029, via bootstrapping, see Materials and methods). Ideal observer decoding showed the same difference between positive deviant encoding in L4, and heterogeneous encoding in L2/3 (*Figure 9—figure supplement 2*). The rate differences for deviants in L2/3 for small stimulus amplitude deviations (±15%) corresponds to a median difference of spike counts of ≥~0.03 spikes per deflection per neuron between the deviant categories. For a 700 ms stimulus at 10 Hz, assuming 300 responsive neurons in the aligned somatotopic representation, this change in rates corresponds to a difference

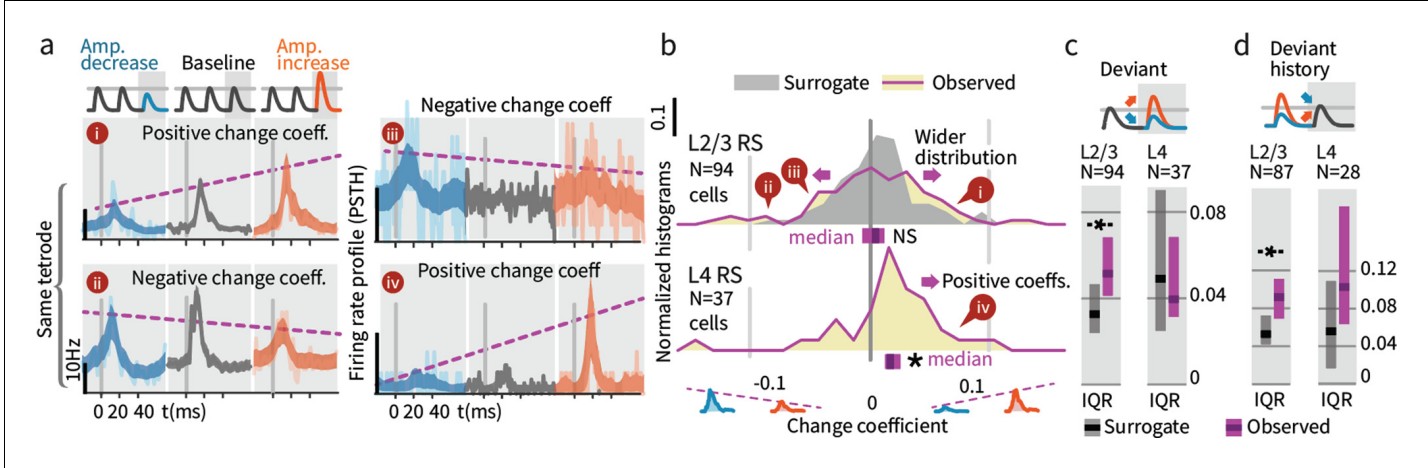

**Figure 7.** Small stimulus deviations were represented by distinct patterns of rate changes in layers 4 and 2/3. (a) Stimulus amplitude deviants were presented after 2–6 repeated baseline amplitude stimulations. Examples show sensory-driven PSTHs, shaded regions indicate the 95% CI. *Red traces* are responses to amplitude increases, and *blue* to amplitude decreases, *gray* are baseline stimuli. We calculated a 'change coefficient', defined as spike count differences between responses to increased versus decreased amplitudes (*purple dashed line*). Neurons had diverse responses, including positive change coefficients (higher firing probability for stimulus increases - i, iv) and negative change coefficients (higher probability for stimulus decreases - ii, iii). (b) Histograms of the distribution of change coefficients for all stimulus-driven RS in L4 and L2/3. The L4 population responded to stimulus changes with positive change coefficients. In contrast, L2/3 showed several types of responses to deviants, with both positive and negative change coefficients, reflected in a broadening of the change coefficient distribution (*purple*) relative to a surrogate distribution of shuffled baseline stimuli (*gray*). Bars show the 95% CIs (*purple*) and the median value. (c) The L2/3 population, but not L4, was broader than a surrogate distribution, quantified via the interquartile range (IQR), ranksum test, bar graphs show median and 95% CI via bootstrap of median. (d) Heterogeneous encoding (reflected in broadening of the distribution) was also observed in L2/3 when initial baseline amplitudes were larger or smaller than a deviant that always had the same intermediate amplitude, demonstrating encoding of deviation from stimulus history and not absolute amplitude encoding in L2/3. The online version of this article includes the following source data and figure supplement(s) for figure 7:

**Source data 1.** Statistics on effect of weak L6 drive on stimulus change representation.
**Figure supplement 1.** Method for matching positions in stimulus trains and for analysis of encoding of stimulus history.

of ~70 spikes per vibrissa, or 250–1000 spikes extrapolated across a typical bout of whisking in sensory decision making such as Gap Crossing.

These heterogeneous responses observed in L2/3 could represent tuning for specific patterns of deviations relative to baseline stimuli. However, the higher firing rates we observed for smaller amplitude stimuli are also consistent with individual cells tuned to specific stimulus amplitudes (*Garion et al., 2014*). To disambiguate these possibilities and test whether L2/3 encoded a true history dependent deviant signal, we analyzed trials where deviant stimulus amplitudes were matched, but deviants were preceded by higher or lower amplitude baseline stimuli (55 sessions, *Figure 7—figure supplement 1*). Neurons with specific amplitude tuning should show a narrowing of the population distribution around zero change coefficient (as the deviant amplitude itself was constant), whereas encoding of stimulus changes should be reflected in a broadening of the distribution. With fixed deviant amplitude, L2/3 RS continued to represent change from stimulus history with a broadened distribution (*Figure 7d*; IQR p=0.004, entropy p=0.014, N = 87), showing that L2/3 RS cells encode history dependent heterogeneous change signals.

In sum, L4 neurons encode deviant amplitude, reflected in a positive correlation between their rates and the direction of amplitude change. In contrast, L2/3 neurons showed a variety of responses to deviants, with different receptive fields for specific patterns of baseline and deviant (*Figure 7*). In contrast to prior reports of increased neocortical excitability with stimulus tuning deviations (*Movshon and Lennie, 1979*), neither population showed significantly increased overall rates for deviants (ignoring deviant direction) (*Figure 6*).

## Layer 6 neurons encoded stimulus amplitude
The low sampling rates of sensory-responsive L6 neurons with extracellular recordings did not allow temporally specific analysis of their tuning for deviance, as performed on single unit firing in L4 and

L2/3. However, the 2-photon imaging data did allow us to ask whether L6 neurons encoded stimulus amplitude in general, and how this encoding might be impacted by the weak optogenetic drive that removed the behavioral impact of a small stimulus deviant (*Figures 2* and *3*).

To this end, we analyzed integrated activity of L6 CT cells by quantifying responses in a 0–2 s window post vibrissa-stimulus offset. Due to the slow timescale of the GCaMP6s indicator, we did not analyze responses to individual deflections. Consistent with electrophysiological data, L6 activity was sparse, as only ~13% of L6 neurons were stimulus-driven (469/3408 cells, *Figure 5—figure supplement 2*). When stimuli without deviants were presented, L6 CT cells encoded baseline amplitude, with significantly higher integrated calcium signals for larger stimuli (p<0.001 signed rank, N = 346, *Figure 8*). To test whether this encoding of amplitude persisted after stimulus adaptation, and therefore could contribute to change representation, we presented stimuli with amplitude deviations after 400 ms. As with baseline amplitude variations, L6 CT also encoded these amplitude differences (p<0.001, *Figure 8*). The timescale of these data, collected at 5 Hz, did not allow disambiguation between explicit encoding of the current deviant, or a delayed encoding that reacts to stimulus changes over timescales >100 ms. Nevertheless, these data show that L6 CT encoded stimulus amplitude as a variation in relative firing rate, which could serve as a baseline for a change-detection computation.

## Weak drive of L6 CT reduced their stimulus amplitude encoding

Given that weak drive of L6 CT cells did not affect baseline or sensory-driven RS firing rates across layers (*Figure 4*), but changed the ensemble of stimulus active L6 CT cells (*Figure 5*) and selectively suppressed the behavioral impact of deviants (*Figures 2* and *3*), we next asked how this manipulation affected change encoding of amplitude in L6 CT. We found that weak optogenetic drive removed the encoding of stimulus amplitude (*Figure 9d,e*, p<0.05, ranksum, evoked ΔF/F change

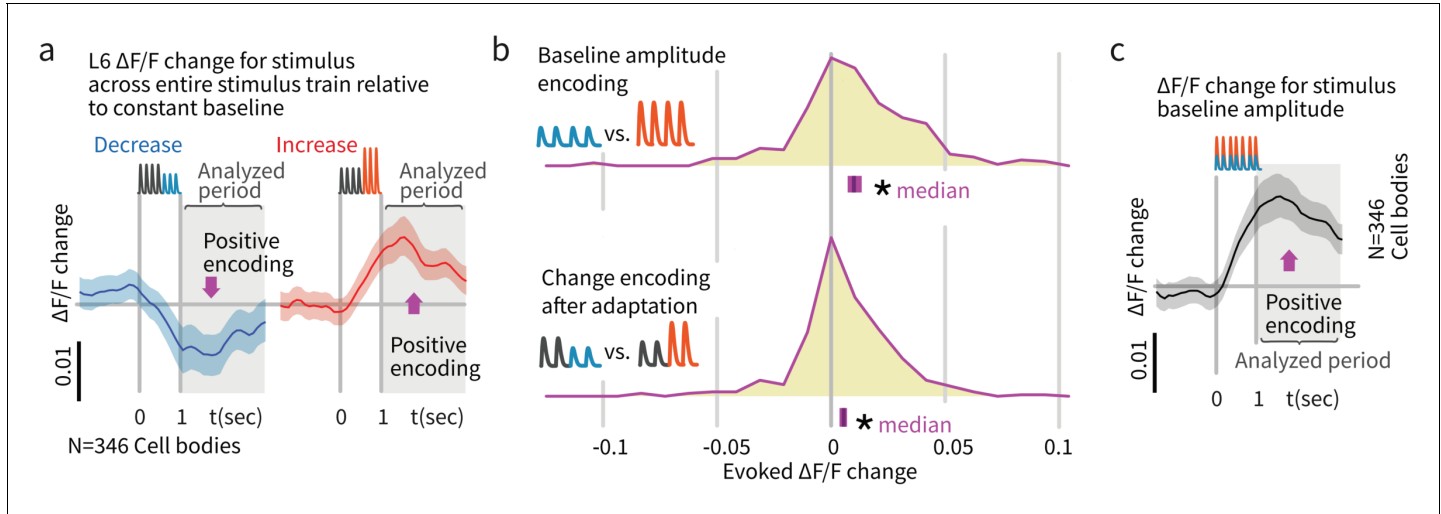

**Figure 8.** Layer 6 pyramidal cells encoded stimulus amplitude. (a) Activity in L6 neurons positively encoded stimulus amplitude, as reflected by larger signals among responsive neurons when the response to larger amplitude baseline stimuli was subtracted from the response to smaller baseline stimuli (N = 346 stimulus responsive neurons out of 2685 imaged). This finding contrasts with the encoding in L2/3 and 4, where we did not observe significant encoding of the baseline amplitude of the stimulus train (*Figure 6*). (b) L6 amplitude encoding was also observed when stimulus amplitude variation was provided by deviants later in the stimulus train. *Blue/red* indicate response to increases versus decreases in deviant amplitude relative to baseline. Because calcium imaging data were acquired at ~5 Hz, no attempt was made to quantify change onsets and only the post-stimulus window was analyzed. Bars show the 95% CIs (*purple*) and the median value. (c) Population data and summary statistics for L6 CT amplitude and change encoding. The online version of this article includes the following source data and figure supplement(s) for figure 8:

**Figure supplement 1.** Weak optogenetic L6 CT drive with ChR2 disrupts encoding of small stimulus amplitude changes and deviants, quantified via raw ΔF/F.

**Figure supplement 1—source data 1.** L6 stimulus and deviant encoding in control and weak L6 drive conditions.

**Figure supplement 2.** Weak optogenetic L6 CT drive with ChR2 disrupts encoding of small stimulus amplitude changes and deviants, quantified via leave-one-out cross-validation for stimulus driven vs. non-driven cells.

between stimuli, control vs. laser). This effect could be explained by a 'shuffling' of stimulus-driven cells, making otherwise non-driven (and poorly tuned) cells vibrissa-responsive and vice versa. Consistent with this hypothesis, cells that were selected to be sensory responsive in the control condition showed decreased average responses in the laser condition (p<0.05, controlled for regression to mean). However, encoding was affected even in L6 CT that were stimulus-driven in the control condition and remained so in the laser condition (p=0.044 rank sum across cells, p<0.0001 across trials, *Figure 8—figure supplement 1*), and the same effect was observed in a cross-validated analysis that classified cells as stimulus-driven and analyzed them in separate trials (*Figure 8—figure supplement 2*). In sum, both the identity of the sensory-driven ensemble and amplitude encoding by L6 CT was disrupted by weak optogenetic drive, despite no change in the net activity in these neurons.

### Weak L6 drive removed deviance encoding in L2/3

We next examined whether weak optogenetic drive of L6 impacted stimulus representation in other layers. We found that change representation in L2/3 was disrupted (*Figure 9b*, p=0.010 entropy reduction, p=0.020 IQR reduction, p=0.008 paired left tailed IQR, N = 94). During optogenetic drive, L2/3 neurons came to represent current stimulus amplitudes with positive change coefficients (CI: [0.002, 0.009, 0.016], p=0.003 signed rank) and reduced their history-dependence (*Figure 9*, p=0.014 entropy reduction, p=0.004 IQR reduction, p=0.043 paired left tailed IQR). *Figure 9—figure supplement 1* shows this same effect quantified via GLM. Shuffling the active L6 ensemble therefore removed the change-specific receptive fields in L2/3, causing them to instead encode current stimuli, making them more similar to L4 cells. These results suggest that stimulus encoding in L6

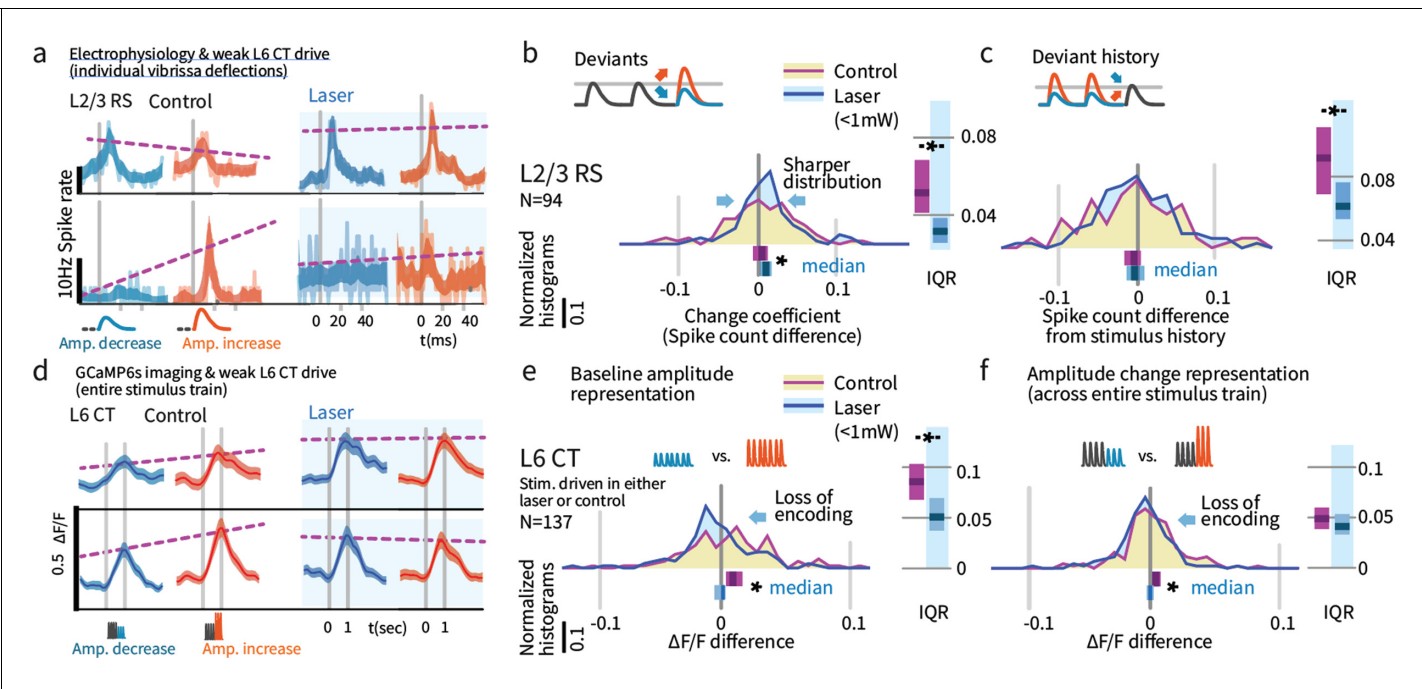

**Figure 9.** Weak L6 drive disrupts stimulus encoding in L6 and the emergence of deviant encoding in L2/3. (**a**) Examples of L2/3 deviant encoding that were reduced with L6 optogenetic drive, of positive and negative sign. (**b**) Across the L2/3 population, weak L6 CT drive caused a loss in the heterogeneity of L2/3 encoding of deviants, reflected as a sharpening of the distribution and a bias to positive change coefficients, paralleling L4 encoding. Bar graphs show the 95% CI and the median value for the distribution, and for its interquartile range (IQR), via bootstrap. (**c**) Encoding of stimulus history in L2/3 was similarly disrupted. (**d**) Examples of L6 CT sensory responses (via Ca²⁺ imaging, see *Figures 5* and *8*) with and without optogenetic stimulation. In these examples, weak L6 CT optogenetic drive disrupted the representation of stimulus amplitude. (**e**) Population distributions show the loss of baseline amplitude encoding in L6 CT, and (**f**) loss of amplitude encoding for changes within the stimulus train. The online version of this article includes the following figure supplement(s) for figure 9:

**Figure supplement 1.** GLM coefficient statistics across layers and conditions.

**Figure supplement 2.** L2/3 RS cells encode stimulus amplitude deviant identity heterogeneously, in contrast to L4.

**Figure supplement 3.** Thalamic relay cells do not encode stimulus deviants as significantly as cortical L4 RS cells.

is necessary for the observed deviance-specific coding to arise in L2/3 through a currently unknown cortical circuit.

Recent studies have concluded that optogenetic drive of L6 impacts sensory responses in visual neocortex through an intracortical pathway, and not by modulation of thalamic relay neurons (*Olsen et al., 2012*; *Bortone et al., 2014*). We tested whether weak optogenetic stimulation affected lemniscal neurons using chronic recordings from ventral posterior medial nucleus in awake animals (Materials and methods as in *Figures 4*, *6*, *7* and *9*). Of 240 well-isolated thalamic single units (N = 3 mice), 25 cells were phasically responsive at short latencies to vibrissal stimulation. These neurons showed weak rate increases with L6 optogenetic drive, in contrast to the suppression observed in prior studies using stronger L6 CT drive (*Denman and Contreras, 2015*; *Bortone et al., 2014*). However, no significant encoding of deviants was observed, nor was modulation of encoding of deviants by L6 drive observed (*Figure 9—figure supplement 3*). While mechanisms that are not captured by these recordings (e.g. changes in thalamic synchrony) could contribute to the loss of heterogeneous encoding in L2/3 with weak L6 drive, rate changes in thalamic responses were not evident. Further, as described above, layers that receive direct lemniscal thalamic input (L4, L5, and L6) did not show explicit deviance encoding other than encoding of current stimulus amplitude. These two forms of evidence, and prior studies finding a direct intracortical pathway as the primary route of influence by L6 CT (*Bortone et al., 2014*), supports a local neocortical mechanism for the emergence of L2/3 deviant responses and in their modulation under L6 optogenetic drive.

In sum, the weak optogenetic drive of L6 CT employed here did not drive changes in overall RS firing rates (measured extracellularly, *Figure 4*, and via Ca2+ imaging, *Figure 5*, *Figure 8—figure supplement 1*), in contrast to the suppression observed when using strong optical stimuli in our own data (*Figures 1* and *4*) and in prior studies (*Olsen et al., 2012*). However, this weak optogenetic stimulus regime did substantially alter sensory encoding by L6 CT. These neurons lost their amplitude encoding, and many individual neurons showed changes in their sensory responsiveness, shifting the specific ensemble of L6 CT activated (*Figure 5*). Further, L2/3 neurons lost their emergent heterogeneous encoding of deviants, showing instead the positive change encoding observed in L4 under normal conditions (*Figure 9*). Weak L6 drive also selectively suppressed indicators of deviant processing in head-fixed (*Figure 3*) and free (*Figures 1* and *2*) behaviors.

## Discussion

We observed layer-specific encoding of deviants and robust behavioral sensitivity to small stimulus deviations. Small stimulus amplitude deviants were correlated positively with firing rates in L4 and L6, with deviant amplitude increases driving higher firing rates. In contrast, heterogeneous encoding, where cells could, for example, encode amplitude increases by decreases in rate, and encoding of stimulus history, emerged in L2/3 neurons. Weak optogenetic drive changed the sensory-responsive L6 ensemble, by facilitating and suppressing different subsets of cells without changing the overall firing rates and reduced their information content about the stimulus. This manipulation also removed the encoding of stimulus deviants in L2/3. Detection of small stimulus deviations in the Gap Crossing with Deviants task, and the benefit of deviants for basic detection, were also lost with weak L6 optogenetic drive without altering basic task performance (*Figure 10*). These results indicate that stimulus encoding by sparse ensembles in L6 contributes to the neocortical circuit that processes sensory deviation but is not required for basic sensory function.

While both receptive fields and behavior were altered by the weak optogenetic drive employed here, we found that this manipulation had no significant effect on RS firing rates in L6 or other layers. The manipulation also had no effect on free Gap Crossing (*Figures 1* and *2*). In contrast, stronger optogenetic drive of these same neurons led to suppressive gain modulation in neocortical response amplitudes (*Olsen et al., 2012*) and disrupted baseline sensory sensitivity (*Figure 1,bc*). The specific impact of the weak L6 manipulation on deviant encoding but not sensory gain (*Figures 4* and *9*), and on deviant-driven sensory behaviors but not basic performance (*Figures 2* and *3*), indicates that this intervention isolated a network mechanism or computation that is selectively involved in stimulus change processing, but not in processing of repeating, or predictable stimuli. The failure of weak L6 drive to impact baseline behavior is in contrast with several findings showing that relatively subtle optogenetic manipulation in SI, for example induced via similarly weak drive of PV+ interneurons (*Sachidhanandam et al., 2013*; *Siegle et al., 2014*; *Lee et al., 2012*) or L4 stellate neurons

**Figure 10.** Summary of main findings. The effects of strong vs. weak drive of L6 CT neurons on behavior and on the electrophysiological signatures of baseline and deviant stimuli.

(*O'Connor et al., 2013*; *Sofroniew et al., 2015*), or direct stimulation of single neurons in other layers (*Houweling and Brecht, 2008*), can affect baseline sensory detection behavior.

Studies of sensory deviation typically manipulate stimulus features such as tone pitch (*Ulanovsky et al., 2003*; *Taaseh et al., 2011*; *Ulanovsky et al., 2004*), for which there are tuned populations, and observe higher neocortical firing rates for deviants, likely reflecting recruitment of new pools of differently-tuned neurons. Here, we specifically sought to minimize such stimulus-specific adaptation. In natural perception, relevant stimulus changes could lack feature changes for which there are such populations, for example decreases in the amplitude of a stimulus (*Voigts et al., 2015*), or higher-order features relayed from other cortical regions. Further, this stimulus design avoids biasing encoding by increases in neocortical drive across all deviants. This lack of stimulus-specific adaptation is evident in the lack of overall increase in firing rates for deviants (*Figure 6e*).

The L6 CT cells imaged and controlled in the present experiments may affect cortical activity via the recruitment of local (*Zhang and Deschênes, 1997*) and trans-laminar (*Bortone et al., 2014*) FS mediated inhibition, and through corticothalamic effects (*Thomson, 2010*; *Denman and Contreras, 2015*). The mechanisms by which specific ensembles of L6 cells influence state or stimulus encoding in superficial layers are still largely unknown, and could also be mediated through a variety of intermediate cell types, layers, and brain areas that were not recorded in the present set of experiments. While the present results directly support a specific role for L6 in the behavioral benefit of deviants, and in the representation of deviation across neocortical layers, further studies are required to determine the circuit, synaptic and cellular mechanisms by which L6 affects neural encoding and behavior.

The encoding of small stimulus changes in L2/3 RS, specifically the history dependence of these responses, is analogous to the emergence of complex temporospatial receptive fields (*Martin and Whitteridge, 1984*; *Gilbert, 1977*) and mismatch encoding (*Zmarz and Keller, 2016*) in upper layers of visual neocortex. The diversity in L2/3 responses, where both positive and negative change coefficients were observed, could reflect neuron types defined by biophysical characteristics (*Jouhanneau et al., 2014*) or projection targets, such as targeting higher somatosensory or motor cortices (*Sato and Svoboda, 2010*; *Chen et al., 2013b*), or may emerge from differential afferent connectivity.

The sparse stimulus encoding observed in L6 CT could represent either an explicit per-deflection deviant encoding, as in L4, or a delayed stimulus or expectation encoding that emerges over timescales greater than 100 ms. The sparsity of L6 CT activity, and their targeting by long-range corticocortical afferents (*Vélez-Fort et al., 2014*), suggests that they might be gated by inputs from other higher-order neocortical areas, as previously proposed (*Lee et al., 2008*). Measurements of rapid responses to individual stimuli with either faster imaging or more comprehensive

electrophysiological methods, and more specific network level manipulations than employed here, will be required to disambiguate these hypotheses.

Even though weak L6 CT drive did not alter stimulus-driven firing rates, this manipulation changed the ensemble of sensory-driven neurons. This finding suggests that a sparse pattern of activity, comprised of both active and inactive neurons in L6, with specific connectivity, is required for deviance encoding. There are two types of mechanisms by which this manipulation could lead to the observed disruption of change encoding in L2/3. The specific population of L6 cells active during optogenetic drive was different from the one active in control conditions. This 'shuffling' alone could lead to a disruption of stimulus representation in L2/3 because the new set of active neurons would be decoded differently by recipient layers. Additionally, we found that the population of L6 CT cells that is active during optogenetic drive carries less stimulus information than the population that is active in the control condition (*Figure 8—figure supplements 1* and *2*), suggesting that any decoding of stimulus information from L6 would be impaired under the optogenetic drive condition. In addition to the set of active neurons, the set of cells that remain inactive, or are suppressed by the vibrissa stimulus across these conditions, could also play a role in the downstream decoding. Regardless of mechanism, our findings do not provide evidence for or against a specific circuit or circuits for change detection, but rather that specific activity patterns in L6 are required for the cortical architecture to perform change detection. Future studies that selectively change the ensemble, for example by specific optogenetic control, will be able to causally test the relationship between the encoding properties of L6 cells and their causal roles.

In this study, we employed two kinds of deviants. Variations from repeating baseline stimuli applied to the vibrissae, and ethologically relevant deviations in the position of a platform in the middle of sampling by freely behaving animals. In the latter case, the stimulus statistics (mainly vibrissa identity, angles, and curvature upon touch) change continuously as the mice approach or retreat from the target platform (*Voigts et al., 2008*; *Figure 2—figure supplement 2*). In both cases, predictive models were at some level formed within the system, driving the change in response patterns and behavior. The similar findings across these paradigms suggests that the mechanism underlying the observed effects could be involved in more general predictive models. That said, during active sensing likely all transitions in perceptual input occur against the backdrop of a working internal model, and distinctions between anticipated and perceived input may be amplified, including the onset of a stimulus following a pause in stimulation. How the present findings relate to dynamics in more naturalistic environmental statistics is an important question for further study.

In sum, we found that stimulus encoding by specific ensembles of L6 cells is required for change encoding in L2/3 and for change-detection behavior, but not basic detection performance. These L6 CT could therefore be one node of the larger laminar cortical circuit for processing of higher-order stimulus features, stimulus context, or expectations reliant on top-down signaling, in agreement with L6 CT targeting by long-range cortico-cortical inputs (*Zhang et al., 2014*; *Vélez-Fort et al., 2014*; *Vélez-Fort et al., 2018*).

## Materials and methods

### Animal subjects

NTSR1-Cre mice (strain B6.FVB(Cg)-Tg(Ntsr1-cre)GN220Gsat/Mmcd, stock number 030648-UCD) (*Gong et al., 2007*) of either sex were used. For some experiments, NTSR1-Cre mice were crossed with a floxed ChR2 reporter line (strain B6;129S-*Gt(ROSA)26Sortm32(CAG-COP4\*H134R/EYFP)Hze*/J, stock number 012569). For head-fixed behavior, one NTSR1-Cre mouse using viral injection, and three reporter line crosses (NTSR1/ChR2 +/+) were used. For Gap Crossing, 6 NTSR1 mice (2 ChR2 viral injection, four reporter line crosses) were used. For electrophysiology, 5 mice of either sex, 4 NTSR1-Cre mice using viral injections and one reporter line cross was used. For 2-photon imaging, 6 NTSR-1 Cre mice with viral injections were used.

### Viral injection

For virus mediated ChR2 expression, we targeted the caudal region of the barrel field (1.5 mm posterior to bregma and 3.5 mm lateral to the midline). Injections were performed through a burr-hole

with a glass micropipette (pulled and beveled, tip diameter of 20–35 μm) attached to a stereotaxic-mountable syringe pump (QSI Stoelting). 300 nl of virus (AAV DIO ChR2-mCherry;~2 × 1012 viral molecules per ml) was injected at 0.05 μl/min at ~800 μm below the dura. All experiments requiring viral transfection were performed >4 weeks after injection. For two- photon imaging,~300 nl of AAV2/1-hSyn-Flex-GCaMP6s (HHMI/Janelia Farm, GENIE Project; ~1 × 1013 viral molecules per ml) (*Chen et al., 2013a*), or in a subset of mice a 1:1 mixture of floxed GCaMP6 and floxed ChR2, all produced by the U. Penn Vector Core, was injected at ~750 μm targeting the posterior c-row barrels, identified by vascular landmarks and confirmed using intrinsic imaging to restrict expression to NTSR1+ neurons. Mice were tested for aberrant expression outside of L6 CT cells either by histology (for behavior and electrophysiology), or by collecting z-stacks (for 2-photon imaging). Mice with fluorescent non-L6 cells were excluded from the study.

## Surgical procedures

Mice were 8–14 weeks old at the time of surgery. Animals were individually housed and maintained on a 12 hr reversed cycle. All procedures and animal care protocols conformed to guidelines established by the National Institutes of Health, and approved by Brown University's Institutional Animal Care and Use Committee. Mice were anesthetized with isofluorane (2% induction, 0.75–1.25% maintenance in 1 l/min oxygen) and secured in a stereotaxic apparatus. The scalp was shaved, wiped with hair-removal cream and cleaned with iodine solution and alcohol. After intraperitoneal (IP) injection of Buprenorphine (0.1 mg/kg) and dexamethasone (4 mg/kg) and local application of lidocaine, the skull was exposed. For some mice, AAV was injected as described. The skull was cleaned with ethanol, and a base of adhesive luting cement (C and B Metabond) was applied. A 0.5 mm diameter area of the skull over left primary somatosensory neocortex was thinned. A stub of fiber-optic cable (0.22 NA, inner diameter of 200 μm, 1.25 mm OD metal ferrule) was glued into place at the side of the craniotomy using transparent luting cement. Custom head posts (*Voigts et al., 2020*; http://www.github.com/open-ephys/headposts_etc) were affixed with luting cement, the incision was closed with VetBond (3M), and mice were removed from isoflurane. Mice were given 3–10 d to recover before the start of water restriction. For electrophysiology, we implanted flexDrives (*Voigts et al., 2013*) with 16 stereotrodes (N = 2, 17 sessions), or tetrodes (N = 3, 58 sessions) made from 12.5 μm polyimide-coated nichrome wire (Kanthal), twisted, heated and gold-plated to 200–400 kΩ impedance. Lateral electrode spacing was 250 μm. Two stainless-steel screws were implanted anterior to bregma to serve as ground. For some mice we injected AAV as described. A craniotomy was drilled over left SI (~1.5 mm posterior, 3.5 mm lateral,~2.5 mm diameter). A fiberoptic stub was added as for behavioral testing, and a large durotomy was opened. A layer of bacteriostatic surgical lubricant was added, and the drive was lowered at an angle of ~15° and fixed in place using dental cement. After recovery (>3 d), mice were habituated to the setup and electrodes were lowered into the brain (~2 hr between individual electrodes) while noting when each electrode penetrated the brain. Mice were water restricted as described. During the experimental life time of mice, electrodes were advanced to target neocortical layers and maintain recording quality (*Voigts et al., 2013*). For 2-photon imaging, titanium headposts were used, the skull around SI was thinned and flattened. A 3 mm craniotomy was made, virus was injected, and a cranial window (*Andermann et al., 2011*; *Goldey et al., 2014*) 'plug' was made by stacking two 3 mm coverslips (Deckgläser, #0 thickness (~0.1 mm); Warner; CS-3R) under a 5 mm coverslip (Warner; CS-5R), using optical adhesive (Norland Optical #71). The plug was inserted into the craniotomy and the edges of the larger glass and animals that did not consume 1 ml of water/session or lost weight were supplemented with water in their home cage several hours after the experiment finished.

## Head-fixed behavioral training and stimulus design

Training began >10 d after postoperative recovery and at least >7 d after onset of water restriction (1 ml/d). Mice were secured to the head-post apparatus and rested on a platform. Initial training procedure was as described before (*Siegle et al., 2014*). White noise (~65 dB) was used to mask auditory cues. If mice licked up to 800 ms after the onset of the vibrissae stimulus, water was delivered. There was an additional time-out period of 2 s for false alarms, and a pre-stimulus delay period (1–4 s) was gradually introduced, during which licking resulted in a reset of the delay timer. Vibrissae were stimulated with a custom stimulator based on piezoelectric wafers (Noliac CMBP09).

Stimulations consisted of deflections with a fast onset velocity and a slower ~80 ms return to baseline with a small (~10% of peak amplitude) negative deceleration period to reduce a 2nd deceleration peak and to reduce the impact of piezo hysteresis. Several vibrissae, centered around the C2 vibrissa, were gripped ~5 mm from the mystacial pad. Amplitudes were calibrated using videography. Piezo elements were replaced if ringing exceeded 10% of the peak amplitude, or if the stimulus amplitude deviated by >5%, or if any hysteresis was measured. Water delivery was controlled by a solenoid valve (Lee Co.), calibrated to give an ~8 µl per opening (30–60 ms). Licking was detected via infrared detectors. After reaching criterion, optogenetic stimulation was added on half of trials. Stimulation started at a variable offset of 0.2–1.5 s preceding tactile stimulation and persisted for the duration of the stimulus. Laser power was ramped up with a gaussian onset profile lasting ~200 ms (*Figure 4*). Vibrissa-stimulus amplitudes were drawn uniformly from a range (~=0–30 mm/sec), adjusted manually to maintain performance while probing small stimulus amplitudes. In 10% of trials, maximum amplitude stimuli were delivered. Behavior depended on vibrissa stimulation and was independent of auditory or visual cues (*Figure 3—figure supplement 1*) Behavioral experiments were controlled using a custom state machine (*Buschman and Nowak, 2010*; http://www.github. com/open-ephys/behavioral_state_machine) written in Matlab via PCI DIO boards (National Instruments). Mice were weighted daily, and animals that did not consume 1 ml of water/session or lost weight were supplemented with water in their home cage several hours after the experiment finished.

## Head-fixed behavioral analysis

Data analysis was performed in Matlab (Mathworks) as described before *Siegle et al., 2014*. Trials were selected based on a d' threshold of 1.2. We also excluded mice in which the false-positive rate was increased in the laser condition (>95% binomial confidence bound). One mouse was excluded based on this criterion, which was established before the start of experimentation. To account for differences in the clamping distances from the follicle, angles and number of vibrissae across sessions, we estimated the threshold amplitude per session: (i) The d' was computed as described and trials with d'>0.8 were analyzed. (ii) A cumulative gaussian was fit to the stimulus amplitude and hit rate, and the median point of the curve was defined as the threshold amplitude for that session. Subsequent analyses were performed on amplitudes normalized to this threshold. This normalization resulted in a hit rate of ~80–100% for stimuli of normalized amplitude 1 in the d'>1.2 filtered data. Trials with responses within 50 ms of stimulus onset as were trials with stimulus deviations later than 400 ms, to exclude licking that was not elicited by the stimulus, or trials in which the stimulus deviation was too late to contribute to the detection performance (reaction times for smallest stimulus amplitudes were ~200–500 ms 95% CI). Binomial confidence bounds (*Figure 3*) were computed with the Clopper-Pearson method at the 95% level. Statistical significance of comparisons between hit rates across conditions was calculated using a bootstrap (10000 samples) on binomial distributions. Mice were performed at chance level when the stimulator was detached from the vibrissa (*Figure 3—figure supplement 1*).

## Experimental design for electrophysiology and 2-photon imaging

Mice rested on a styrofoam ball supported by an air cushion. Mice were water restricted and monitored as described, and licked a spout to indicate stimulus detection for reward, but no time-outs, or catch trials were used. Sessions were stopped at signs of animal distress and session durations were increased over the first 2–3 weeks of acclimatization, resulting in sessions of ~2000 trials over ~2 hr. Stimuli were delivered as described, with deviants of relative amplitude of the deflections between ±10% and±15%. Amplitudes were calibrated to the range that correspond to ~80–100% hit rate in the behavioral detection task. The interval between stimuli was 3–5 s. For 2-photon imaging, mice were not water restricted and stimulus amplitudes were sampled from two baseline stimuli and two deviant conditions (increase to 120% or decrease to 80%) in order to increase statistical power for the sparse L6 responses.

## Gap Crossing behavior

Mice (N = 6) were implanted with plastic head-posts and fiber stubs, and water restricted as described, vibrissae on the side ipsilateral to the fiber were trimmed. Two mice also had implants for

electrophysiology. The gap crossing apparatus consisted of two facing platforms (*Hutson and Masterton, 1986*) (58 mm wide) over a custom LED backlight (650 nm). Mice were habituated for 2 days prior to the experiments. On day 3, the optical fiber was attached and masking noise (~80 dB) was introduced. After mice crossed the gap in either direction, a water reward (~0.01–0.05 ml) was delivered manually, and a new platform position was chosen between 45 and 65 mm. In half the trials the laser was on for >1 s prior till ~2–3 s after the crossing. In a subset of trials, one platform was retracted by 2 mm within 8 ms via a voice coil actuator (*Voigts et al., 2015* and *Figure 2*, *Figure 1—figure supplement 1*, and *Video 1*) while the mouse was palpating it. Mice were run every other or third day, sessions ended when mice either lost interest in crossing, fell from the platform, or tangled the optical tether. The gap was filmed from above at 315 Hz (Pike 032B, Allied Vision Technologies). In control sessions, the optical cable was attached to a mock ferrule that directed the light to a position rostral of the actual fiber stub implant.

## Gap Crossing analysis

The mouse nose distance to the target platform was tracked using custom scripts in Matlab. Trials were identified as attempted (mouse reached over the gap) or completed crossings. For analysis of sensory disruption using high laser powers, the probability of crossing was computed from all trials with gap distance <6 cm. For other analyses, only trials in which the mice crossed within 5 s were further analyzed. The nose position over time was aligned to the position at which the mouse had committed to a crossing attempt without touching the target yet, extending over the home platform by ~7 mm, corresponding to a position of −20 mm in the imaging reference frame. For analysis of the whisking pattern, subsets of vibrissae were tracked using an automated tracker (*Figure 2—figure supplement 2*, *Voigts, 2020b*; http://www.github.com/jvoigts/whisker_tracking) and the median angle of all tracked vibrissae was analyzed.

## Optogenetic stimulation

In all experiments, light was delivered through a jacketed fiber-optic cable 200 μm in core diameter and 2.5 m long with a numerical aperture of 0.22 (Doric Lenses) connected to a 450 nm diode laser (powertechnology.com) using a collimator (Thorlabs PAF-X-15-PC-A). The fiber was connected to the animal's head via mating metal ferrules in a zirconia sleeve. For head-fixed behavior, ferrules were shielded with black plastic tape and the head of the mouse was illuminated with a blue masking LED that did not illuminate the stimulator or vibrissae. Light loss in the implanted fiber stub was measured for each implant. The amplitude of the light stimulus was calibrated regularly with an optical power meter (Thorlabs PM100D with SI20C sensor) to up to 1 mW at the surface of the skull, resulting in ~0.1–0.5 mW in neocortex (measured through the skull and metabond after perfusion). In a subset of sessions, higher laser power (~2–5 mW, see *Figure 4*, or up to ~10 mW for Gap Crossing, *Figure 1*) was used. The chronic implantation of a optical fiber results in a somewhat decreased power delivery to the brain due to inevitable regrowth of dura under the implantation site. Direct 1:1 comparisons of the light powers of the chronic experiments to the acute experiments where the fiber was placed directly on the brain (*Figure 4a*) are therefore not possible, and it should be assumed that a somewhat higher light power is required in the chronic case to achieve the same extent of optogentic activation as in the acute case.

## Analysis of electrophysiology data – acquisition and pre-processing

Unless indicated, we used non-parametric Wilcoxon rank sum/Mann Whitney U-test tests for comparing groups (non-paired), or Wilcoxon signed rank tests for testing medians versus zero or comparing paired measurements. We used animal cohort sizes consistent with practices in the field (*Figure 1—figure supplement 1*) and recorded the highest possible sample sizes per animals as allowed by methods (N of cells), and animal health, comfort, and behavior (N of sessions, length of recordings). No initial sample size estimation was used, nor were sample sizes increased after analysis if the data. Extracellular voltage traces were band-pass filtered to 1–10000 Hz at acquisition (3[rd] order butterworth filter), and band-pass filtered to 300–9000 Hz (zero-phase acausal FIR) for analysis of spiking. Sessions in which vibrissae had slipped out of the stimulator were excluded. Spikes were sorted into single units using Simple Clust (*Voigts, 2020a*; http://www.github.com/open-ephys/simpleclust). The 90% quantiles of neuron count/session were 17 and 39 over all 75 sessions. We

recorded the depth at which electrodes penetrated the brain, marked by emergence of off-diagonal peak-to-peak amplitudes in the MUA activity (presumably from L1 axons) as the 0 mm position to estimate the depth of electrodes. We combined this information with the drive screw position and angle of the drive to estimate depth. In deeper layers, we additionally used the depth at which electrodes entered the white matter (loss of cortical activity) as a further reference point. The mapping from depth to layers was approximately (in µm): L2/3, 100–350; L4, 350–450; L5, 450–650; L6,>650 but was adjusted to take electrode angle and curvature of neocortex and white matter borders into account. Drive depth estimates were verified at the L3/L4 boundary of primary somatosensory neocortex (SI), via the stimulus-evoked LFP signature (*Castro-Alamancos and Connors, 1996*).

### Analysis of electrophysiology data – classification of sorted units

We classified neurons as regular spiking (RS) and fast-spiking (FS) by spike waveform (*Bortone et al., 2014*). Stimulus-driven neurons were classified by fitting a generalized linear model (*Truccolo et al., 2005*) (GLM) to the PSTH. We classified cells as phasically driven if either of two conditions were met: (i) An offset term and six bins (basis functions) spanning the first 100 ms of the first vibrissa deflection were fit, coefficients for at least two bins were significantly nonzero at a P level of 0.03 and any coefficients other than the offset term had a lower standard error bound >0.002. (ii) A constant term and six repeated bins for over first 100 ms of the first three deflections were fit, these coefficients were shared between deflections capturing cases of weaker but sustained stimulus drive. Additionally, one parameter for each 100 ms vibrissa deflection period after the first one was used to avoid false positives due to slower firing rate drifts. Cells were classified as driven if the coefficients for the first three deflections satisfied the same conditions as in (i). Classifications was verified manually to choose thresholds but no manual corrections were made.

### Analysis of electrophysiology data – population coding analysis

To plot example PSTHs (*Figures 7* and *9*), we computed confidence bounds using a state-space method (*Smith et al., 2010*). These analyses were used for visualization purposes only. The random position of deviant stimuli resulted in more baseline than deviant deflections, and more baseline stimuli early in the train (stimuli after the deviants were not analyzed). For analyses that are susceptible to biases of unequal N and adaptation effects, such as change coefficients, a histogram matching procedure was used to match the number and position in the stimulus train across baseline and deviant stimuli (*Figure 7—figure supplement 1*). The effect of stimulus history on firing rates was analyzed using subsets of trials in which the stimulus amplitudes were matched but were preceded by higher or lower amplitude stimuli by matching the stimulus amplitude distributions (*Figure 7—figure supplement 1*). All statistics of change coefficients were computed as 95% confidence intervals (CI) of the median using a 1000 or 10,000-fold bootstrap. Spread of distributions of change coefficients was quantified as the difference between the observed and a surrogate distribution (computed from position matched, randomly re-sampled baseline stimuli) via the interquartile range (75th−25th percentile) and Shannon entropy (in bits): H(observed) - H(surrogate); H(h)=-sum_i(P(h_i) * log2 P(h_i)). The null distribution is computed by re-sampling trials within-cell and is therefore affected by cell-dependent differences in variability in the same way as the true distribution. 95% confidence bounds and significance levels for these statistics were determined via bootstrap analysis. Entropy was quantified via the difference between pairs of binned distributions, so choice of bin size had no significant effect. Where paired samples per cell were available, as in the effect of the optogenetic manipulation, a bootstrap on the median of the absolute value minus the population median was used: median(absolute(coeff_cell-coeff_population)).

### Analysis of electrophysiology data – per-neuron GLM analysis

We quantified encoding in individual neurons with a GLM. We analyzed parameters for spike count as a function of stimulus deviation (*Figure 9—figure supplement 1*), mirroring the direct computation of change coefficients (*Figure 7*). The features used in the model were stimulus deviation (−1: decreases, +1: increases), baseline amplitude, and spiking history (for seven precedent deflections). The adaptation profile was modeled with a separate feature per deflection, linked with a quadratic penalty term on the pairwise difference (weight 10). A separate quadratic regularization term with (weight 1) penalized large parameters (other than constant) to avoid over-fitting. The regularizing

matrix (q) was: q = 1*I+10*D (I: Identity matrix, D: difference operator). Model parameters (w) were estimated from spike counts (Y) via min_w(-log(p(Y|w)) + 0.5*w'*q'w). 95% confidence bounds were obtained using a 100-fold bootstrap. False-positive rates were calculated by fitting to surrogate data (as described). Control, laser, and deviant conditions (increases/decreases) were fitted independently.

## 2-photon imaging

A two-photon microscope (Bruker/Prairie Technologies) using an 8 kHz resonant galvanometer (CRS) for fast x-axis scanning, and a non-resonant galvanometer (Cambridge 6215) for y-axis increments was used. In some sessions, non-resonant scanning in a smaller imaging window (variable region ~100×80 px) was used. Frames were 512 × 512 pixels (resonant) or smaller (non resonant) and scanned at >5 Hz. Objectives (Nikon 25 × 1.1 NA or Nikon 16 × 0.8 NA) were rotated to the window plane. GCaMP6s was excited by a pre-chirped Ti-Sapphire laser (Spectra Physics; MaiTai) at 980 nm. Power at the sample was 20–60 mW for superficial imaging (<450 μm), 60–80 mW for deep imaging (>450 μm), when scanning at ~5–10 Hz with an approximate pixel dwell time of 1-2μs. Emitted photons were collected through the imaging path to a multi-alkali PMT (Hamamatsu; R3896, digitized with 14-bit resolution). A typical session lasted 2 hr. We found no activity 'run-down', substantial bleaching or cellular damage over the session, consistent with the what other studies using similar laser intensities have reported (*O'Connor et al., 2010*; *Huber et al., 2012*). In about half of implanted animals, we were able to image cell bodies of NTSR1+ layer 6 CT cells (3408 ROIs total) at depths between ~650–800 μm. Good image quality at commonly used excitation laser powers (see above) at these depths was possible likely due to the sparse and relatively localized expression in L6 (approximate diameter of region with cell bodies ~300–400 μm), which results in relatively little fluorophore above the imaging plane, resulting in better signal-to-noise ratios at such depths than would be possible with denser labeling (*Theer et al., 2003*). If the optical quality of the implanted window was non-optimal, due to dura re-growth, animal age or any surgical imperfections, L6 imaging became impossible. All analysis routines were written in MATLAB. Motion artifacts, small movements in the x-y plane were corrected with rigid-body image alignment (*Bonin et al., 2011*) using a DFT based method (*Guizar-Sicairos et al., 2008*) or a similar affine deformation to register to templates averaged from 1000 low-motion frames. To manually identify ROIs, we calculated mean and standard deviation projections, and correlation coefficients for the entire image relative to a seed pixel, and areas of continuous or nearby highly correlated pixels were grouped into the ROI.

## Simultaneous 2-photon imaging and optogenetic stimulation

Light was directed at the entire imaging area from a 200 μm fiber at a ~40 degree angle. To minimize light artifacts and PMT damage, we used a blocking filter (Semrock OD 6, custom NIR block, notches to block 460–470 nm and 560–570 nm). Light from blue (470 nm) or yellow (560 nm) LEDs, driven with a high-speed LED driver (cyclops [*Newman et al., 2015*], designed by Jon Newman, http://www.open-ephys.org/cyclops) was pulsed for 75μs after each 4th or 8th x-scan line. Overall pulse rates were >200 Hz, functionally equivalent to constant light (*Lin et al., 2009*). Light levels were adjusted manually to integrated powers of ~0.1 mW (for ChR2). X-scan lines following laser pulses were brighter due to the light stimulation and were replaced by interpolated data from preceding and following x-scan lines, whether the LED was on or off. Remaining slight image brightening was corrected off-line (see below).

## 2-photon data analysis

Unless indicated, Wilcoxon rank sum or signed rank tests, and bootstrapping for testing IQRs were used, as described for electrophysiological data. Fluorescent values F were extracted from ROIs, the baseline fluorescence F0 was computed as the 30th percentile in a 200 s sliding window and ΔF/F was computed as (F-F0)/F0. Annulus-shaped ROIs were computed to estimate neuropil contamination (*Chen et al., 2013a*; *Bonin et al., 2011*; *Kerlin et al., 2010*) by eroding out 20 pixels in the x-direction from each somatic ROI (this ensures that if there is any specific artifact from the pulsed optogenetic drive, it affects the cell body and neuropil ROIs equally) and excluding other cell bodies from this neuropil ROI (*Bonin et al., 2011*; *Kerlin et al., 2010*). For all analyses of firing rates,

residual image brightening due to light artifacts was corrected by subtracting an average image brightening profile averaged from all neuropil ROIs over the entire session. All other analyses are computed as differences in evoked fluorescence between stimulus conditions within the same cell and laser condition, and were therefore not affected by the light artifact correction. Stimulus-driven ROIs were identified by comparing the 90% quantiles of the $\Delta F/F$ for each ROI in the pre-stimulus period for all stimulus conditions ($-1500$–0 ms) with the 10% quantile in the stimulus period (500–1500 ms), and ROIs with non-overlapping quantiles were analyzed further. Cells were classified either in control trials or optogenetic drive trials, or both. Change coefficients were defined as in the spiking data, but owing to the slow timescale of GCAMP6s, we analyzed the difference of the stimulus-evoked fluorescence between baseline and deviant stimuli over the entire post-deviant stimulus time (0-2ec after stimulus offset) instead of analyzing individual deflections. Control levels were computed as described before.

## Code availability

All custom software used in this study is freely available: Behavioral experiments were controlled using a custom state machine (*Buschman and Nowak, 2010*; http://www.github.com/open-ephys/behavioral_state_machine) written in Matlab via PCI DIO boards (National Instruments). Vibrissae were tracked using an automated tracker (*Figure 2—figure supplement 2*, *Voigts, 2020b*; http://www.github.com/jvoigts/whisker_tracking). Spike sorting was performed using a custom manual sorting tool (*Voigts, 2020a*; http://www.github.com/open-ephys/simpleclust).

## Acknowledgements

Supported by the NIH: R01NS045130 to CIM and F32MH100749 to CAD. We thank HHMI/Janelia Farm Research Campus and their GENIE Program (V Jayaraman, R Kerr, D Kim, L Looger and K Svoboda) for making GCaMP6s available, Tim Buschman for providing behavioral state machine code, and Jon Newman for providing the cyclops LED driver. We thank Scott Cruikshank and Shane Crandall for discussions and comments on the study and manuscript, and Tyler C Brown, Joshua H Siegle and Laura D Lewis for their feedback on the manuscript.

## Additional information

### Funding

| Funder | Grant reference number | Author |
|---|---|---|
| National Institutes of Health | R01NS045130 | Christopher I Moore |
| National Institutes of Health | F32MH100749 | Christopher A Deister |

The funders had no role in study design, data collection and interpretation, or the decision to submit the work for publication.

### Author contributions

Jakob Voigts, Conceptualization, Data curation, Software, Formal analysis, Validation, Investigation, Visualization, Methodology, Writing - original draft, Writing - review and editing; Christopher A Deister, Conceptualization, Resources, Data curation, Software, Formal analysis, Validation, Investigation, Methodology, Writing - review and editing; Christopher I Moore, Conceptualization, Resources, Data curation, Formal analysis, Supervision, Funding acquisition, Validation, Project administration, Writing - review and editing

### Author ORCIDs

Jakob Voigts https://orcid.org/0000-0002-5174-7214
Christopher A Deister https://orcid.org/0000-0002-9579-918X
Christopher I Moore https://orcid.org/0000-0003-4534-1602

## Ethics

Animal experimentation: All procedures and animal care protocols conformed to guidelines established by the National Institutes of Health, and approved by the Institutional Animal Care and Use Committee (IACUC) protocol (#1710000308) at Brown University (PHS Animal Welfare Assurance number D16-00183).

## Decision letter and Author response

Decision letter https://doi.org/10.7554/eLife.48957.sa1
Author response https://doi.org/10.7554/eLife.48957.sa2

## Additional files

### Supplementary files

• Transparent reporting form

### Data availability

Underlying data for all main result figures are included in the supporting files.

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
