## [Decision Letter]

**Acceptance summary:**

In this paper, the authors present an interesting new role for layer 6 in making predictions, specifically in processing deviants from expected sensory inputs. The authors combine advanced recording methods and rigorous statistical analyses to quantify neural and behavioral responses to small stimulus deviations. They show that slight modulation of the sparsely active ensemble of L6 neurons by optogenetics has no impact on basic sensory function, but blocks deviance encoding in L2/3 and changes behavioral responses to deviating stimuli.

**Decision letter after peer review:**

Thank you for submitting your article "Layer 6 ensembles can selectively regulate the behavioral impact and layer-specific representation of sensory deviants" for consideration by *eLife*. Your article has been reviewed by three peer reviewers, one of whom is a member of our Board of Reviewing Editors, and the evaluation has been overseen Laura Colgin as the Senior Editor. The following individuals involved in review of your submission have agreed to reveal their identity: Garrett B Stanley (Reviewer #2); Jeanne Paz (Reviewer #3).

The reviewers have discussed the reviews with one another and the Reviewing Editor has drafted this decision to help you prepare a revised submission.

Summary:

In this paper, the authors present an interesting new role for layer 6 pyramidal neurons as encoders of stimulus amplitude that is used to encode subtle deviation from expected sensory input within the cortical column. The authors combine effective sensory stimuli, advanced recording methods, and rigorous statistical analyses to find the sparse 13% of L6 neurons that respond to sensory stimuli and show that L2/3 responds to deviations from expected stimuli. In addition, they show that slight modulation of L6 neuron activity by optogenetics blocks deviance encoding in L2/3, and changes behavioral responses to deviating stimuli.

Essential revisions:

While all three reviewers appreciated the general question asked, the novel approach, and the rigor of the methods, all expressed considerable concern that the conceptual presentation was far from clear. Detailed comments are listed below, but overall reviewers agreed that while further experimentation is unnecessary, better rationalization of approach, clearer connections of subtopics and methods, and explicit discussion of limitations and future directions were necessary.

Before we list the specific critiques, we have included the general comments of each reviewer so that the authors fully appreciate the common concerns:

For each experiment, the authors have chosen the appropriate recording method. Imaging deep to find sensory-responsive L6 neurons, characterize their encoding of amplitude, and then show remapping of the sensory-responsive neurons with weak optogenetic drive was particularly effective. In cases where they make arguments about encoding across the population, e.g. for deviance encoding in L2/3, they thoroughly describe response distributions with appropriate statistics and interpret them honestly in the text.

The weak optogenetic stimulation used to reveal deviance encoding in L6 and L2/3 is a potential concern. It is the key manipulation in the work, and so readers must be thoroughly convinced of its meaning. The authors distinguish this manipulation from stronger drive by comparing sensory-evoked firing rates and sensory detection behavior with weak and strong optogenetic stimulation. Strong optogenetic drive blocks sensory responses across the cortical column and impairs sensory detection behavior as shown before, but weak drive does not appear to impact these measures of baseline sensory encoding (except L4 FS cells). However, authors should mention the decrease in firing rate at laser onset in L2/3 RS, L5 RS, and L4 FS cells, as it reflects an increased change in firing rate in response to the first sensory stimulus.

While it is clever to use this manipulation to show L6's importance to deviance encoding, it is overinterpreted in the discussion. The claim that weak optogenetic drive activates a separate circuit is not supported by the data. The second claim that an ensemble of L6 neurons is used to encode deviance in L2/3 is suggested by the surprising remapping of sensory-responsive L6 neurons, but it is not directly proven. These statements should be toned down to better match the findings in the paper.

The questions being asked are important, the experiments and study novel and really well done, and the findings exciting and likely to spawn a range of follow up questions and studies. I think that the organization of the manuscript is sub-optimal, often making it confusing, and alternately dense vs. vague. Further, there is a somewhat confusing central result – that optogenetic activation (and not inactivation) of L6 seems to disrupt deviance encoding, which in itself is not a problem (most results related to complex circuitry are confusing). But the text asserts that L6 neurons are important for deviance encoding, suggesting that activation of L6 should enhance deviance encoding in some way, which is not the finding. Most importantly, as discussed in more detail below, the central behavioral task does not seem well-suited for the questions being probed (and is essentially abandoned after the first figure – see my recommendations below). The Abstract and Introduction establish an expectation for the reader that a behavioral task targeting this direction should have a very clearly measurable role for a sensory deviant, upon which the rest of the manuscript revolves. Although the behavior used may be somehow indirectly related to this issue, it is quite vague in precisely how, and seems forced into the study. In addition, the rest of the manuscript suffers at times from switching back and forth between different measurement modalities (ephys and 2P imaging) and stimulus design (single deviant, multiple deviants in a row), making it quite challenging as a reader to digest. In the spirit of trying to help the authors sharpen their argument and make the most impact in this really important and prescient issue, the following set of comments are offered:

1) Behavior – The authors make a really nice argument for the potential role of L6 cortical neurons in deviance detection, which motivates the study very well. They then propose an optogenetic manipulation of L6 neurons, in the context of a deviance detection task in mice. However, the behavior task that was implemented in Figure 1 is certainly not the most obvious choice for a deviance detection task, and it is really puzzling why this particular task was chosen (although one may still like the task). If the task is understood correctly, the mouse is trained to whisk across a gap, to contact a platform to which they can cross the gap for a reward. The deviance is created by retracting the platform by 2mm at random times during bouts of tactile sampling. Although they authors demonstrate that this has a demonstrable effect on behavior, in that they slow their approach to cross the gap, it is really unclear how the task might be affected by the deviance, per se, as in the classical implementations of deviance detection. One would have expected a more focused behavioral paradigm, where the effects of the deviance is explicitly measured through standard psychophysical means. It is true that the authors apparently tested this in such a task, as discussed in Supplemental Figure 4, but the main behavioral task of the paper presented in Figure 1 seems not well-aligned with the goals of the manuscript in a way that readers may find confusing. To be more specific, the inputs to the whisker system as the animal is actively palpating the platform (and move their body and head) are not clear, and it is not clear that the movement of the platform would create a deviant input consistent with how this is being described throughout the manuscript.

As a side note, Figure 1 was found to be quite confusing and difficult to understand what was being presented – so in any case, simplifying and clarifying was suggested. But another possibility is to remove this behavior from the manuscript, as it is quite different from the rest of the data/figures. Would there be any way to pull the behavior of Supplemental Figure 4 back into the main manuscript as the central behavior? Again, there was enthusiasm about the overall arc of the story, but concerns were raised about whether the behavior really fits.

2) Overall, the manuscript is somewhat confusing – and part of this is likely due to the organization. One thing that plagues the manuscript is the use of different stimuli for almost every figure and analysis. It starts in Figure 1 with the active palpation in the gap jumping task, switches to 10 Hz train in Figure 2, and then to a 10Hz train with a single deviant in Figure 4 and Figure 5, switches to a 10Hz train with multiple deviants in sequence for the 2 photon imaging (Figure 3 and Figure 6), and finally with a mixture in Figure 7. It is likely the case that the 2P imaging required integrating across multiple stimulus presentations for practical reasons, necessitating this design, but overall, this makes for a pretty confusing manuscript to navigate for the reader, with all of this jumping around between stimulus design and recording modality. A potential concern was raised about the fact that the deviant is comprised of a sequence (and presumably the appearance of the second and third stimulus in the train might be different from the first, which is the true deviant). When is the deviant no longer a deviant, and instead the new background? It was thought that the authors can fix this through writing and organization, but as written, it is quite difficult to navigate. Perhaps a better explanation of the "why" for the differences in the stimulus design? Again, this is all exacerbated by the jarring transition from the behavior in Figure 1 to the rest of the paper as described above in point #1.

3) The central over-arching assertion is that L6 neurons are vital for deviance encoding (and perhaps setting the stage for a predictive coding framework). However, the manipulations that are done are to either selectively engage a sub-population of L6 neurons through weak optogenetic stimulation without changing overall firing rate, or to strongly activate L6 neurons with strong optogenetic drive, both of which seem to “disrupt” deviance encoding. Unless there was a misunderstanding of something here, it would seem that the presence of active L6 neurons disrupts deviance encoding, rather than facilitating it. This itself is interesting, but confusion exists about statements like the last sentence of the Abstract, "These findings indicate that, despite their sparse activity, specific ensembles of stimulus-driven L6 neurons are required to form neocortical predictions, and for their behavioral benefit." The title of the manuscript captures the findings – that L6 regulates this, but the other text seems to suggest that L6 facilitates this, which seems to be the opposite of the findings. If this description somehow misconstrued the results, then the authors should find a way to clarify this text.

4) Confusion exists about the results at the core of the study. Figure 4 says there is no net relationship between the amplitude of the stimulus deviance and the change in firing rate for L4 or L2/3 neurons (Figure 4G), but then Figure 5 dissects this, only to conclude that L4 has deviance encoding (positive correlation of deviance sign and change in firing rate), but not L2/3 (Figure 5B).

5) The confusion experienced in Figure 4 and Figure 5 continued with the optogenetic manipulation of L6 in Figure 7. Why were results not presented for L4, as opposed to just recordings in L6 and L2/3, since Figure 5 shows that the deviance encoding was emergent in L4 and not L2/3? Is it the case that there is just no effect? Furthermore, the title of Figure 7 is "Weak L6 drive disrupts both stimulus encoding in L6 and the emergence of deviant encoding in L2/3", yet the results in Figure 7B seem to suggest that when the laser is turned on to weakly drive L6 neurons, the deviance encoding emerges (as shown by the shift of the median away from zero), rather than being disrupted. It could be the case that this was not interpreted correctly during review, but in any case, readers would also likely struggle with this. Are 7A and 7B consistent?

6) In Figure 2 there are dramatic differences in the quality of recordings in panel a versus b. Was this due to the use of tetrodes in b and silicon probe in a, and if so, is this something we should be worried about for the rest of the study that relies on the laminar recordings from the silicon probe?

7) In Figure 4, the authors show the results of small variations in stimulus amplitude in L2/3 and L4 but not L6. They later show in Figure 6 the results of L6 encoding stimulus amplitude – why did they choose to present these results in separate figures?

8) It is also not clear if the pattern of stimulus trains used in Figure 4 vs Figure 6 are the same or different. For example, in Figure 4 they use constant stim trains of one amplitude vs. another, and constant stim trains with a single deviation. In Figure 6 it looks like they test a constant stim train of one amplitude followed by a constant stim train of another amplitude. Why are these results analyzed differently?

[Editors' note: further revisions were suggested prior to acceptance, as described below.]

Thank you for submitting your article "Layer 6 ensembles can selectively regulate the behavioral impact and layer-specific representation of sensory deviants" for consideration by *eLife*. Your article has been reviewed by two peer reviewers, one of whom is a member of our Board of Reviewing Editors, and the evaluation has been overseen by Laura Colgin as the Senior Editor. The following individuals involved in review of your submission have agreed to reveal their identity: Garrett B Stanley (Reviewer #2).

The reviewers have discussed the reviews with one another and the Reviewing Editor has drafted this decision to help you prepare a revised submission.

Summary:

The paper is much improved in terms of clarity of presentation, and description of the multiple approaches and experimental findings. However, both reviewers have remaining concerns about both presentation and interpretation, and detailed comments from the reviewers are provided below.

Reviewer #1:

The authors have responded to the previous critiques, and the paper is now much more readable. The new figures highlight the exact experimental design and interpretation. Overall, it is much improved.

However, there are still some points that are not clear, and further revision could improve comprehension.

1) Subsection “Weak depolarization of L6 neurons changed the identity of stimulus-driven ensembles in L6 without changing mean firing rates across layers”. I think you need to say something like we tested the effects of "weak and strong stimuli, analogous to those disrupting gap crossing, etc.", on basic sensory encoding, unless you did both sets of experiments in each mouse, in which case you might want to say that explicitly.

2) Figure 2, while improved, is a bit confusing, as it indicates time and space in a complex way. Might it be possible to change the depiction of the moving platform to make it appear discontinuous, perhaps with multiple overlapped shadowed positions of the platform? As depicted, one has the visual impression that a platform with a recessed portion is moving left to right.

Reviewer #2:

In this work, the authors set out to address the functional role of cortical Layer 6 in predictive coding in the context of a deviant detection task in the awake, freely behaving mouse. They specifically put forth the idea that this particular circuit may be involved in predictive coding, where predictions are made based on past experience, and compared with incoming signals, perhaps driving the detection of anything that deviates from the expectation. As with the original manuscript, the general theme of linking L6 CT and predictive coding is very intriguing, and while there are increasing numbers of studies investigating the function of L6 CT neurons due to the new tools enabling the specific targeting of this cell type, the actual role of these neurons in behavior and perception is largely an area of speculation at this point. So, we really need studies like this. Moreover, the authors present very interesting results, such as a change in behavioral performance due to low L6 drive and a complete shutdown of sensory-based performance due to strong L6 drive, and the engagement of a different ensemble of neurons in L6 with weak optogenetic drive of L6 (that was also linked to weaking of behavioral sensitivity to deviant stimuli, but not affecting behavioral performance).

Following the first round of critiques, the authors made substantial changes to address the concerns of the reviewers, making an earnest attempt to clarify and streamline the argument. The revisions conducted by the authors were very helpful in making the objectives and findings much more clear. However, this also highlighted some issues that are still troublesome:

1) With the revision, the description of the task is now much more clear – the breaking of the original Figure 1 into the new Figure 1 and Figure 2 and the corresponding text around this improve the clarity a lot. Furthermore, the behavioral consequences of the L6 CT activation are clear – the movement of the target platform leads to a slow-down in the crossing, and weak ChR2 activation of L6CT neurons restores the behavior to the normal speed (the same as in the absence of the platform move).

What I still find to be a stretch is the connection of this behavioral paradigm to the concept of deviance detection. While the behavioral results are unambiguous – as shown in Figure 2C,E,G, the nature of the sensory (whisker) stimulus during this behavior is still hazy. The figure shows cartoonish representations of the whisker inputs as the mouse actively contacts the platform before and after the movement of the platform, showing a sequence of identical amplitude whisker deflections, followed by a dramatic reduction in the amplitude of the whisker input after the platform moves, which could be thought of as a deviant input, but whether the whisker input actually looks like this or not, is entirely unclear. This may indeed be the case, but I think just suggesting it without showing it, leads to some discomfort about the connection.

I made this comment in the first round, and while I do think the authors took my comment seriously, they seem to fundamentally disagree. So, I would just like to restate this, in the spirit of helping them understand why readers may not easily lock on to this connection. For me, the transition between the first part of the paper and the rest is still quite jarring. Overall, I love the gap-crossing paradigm, and I think there is something important happening here when the hesitation induced by the platform move is removed with L6CT activation, but I can't help but see this as disconnected from the rest of the paper.

2) Overall, this paper is a tough read. I believe this stems from the use of multiple behavioral tasks (Gap Crossing, whisker direction deviance, whisker amplitude deviance), and the jumping around between different cortical layers and measurement techniques. For example, with the behavior, related to point 1 above, Figure 3 discusses a direction deviant task, and the authors attempt to lead the reader to think this is a natural next step following the gap crossing task (saying that the platform move likely changes a lot of things, including amplitude and direction of motion, but none of this is shown). Even if the gap crossing results occur because the first (and maybe subsequent) whisker contacts following the platform move are different (smaller?) than those before the move, it is really hard to see why the direction deviance task would be the logical follow-up to this. As a side note, I'm not really sure what to make of the fact that for the direction deviant task, the hit rate for the deviant is higher than baseline for weak stimuli and lower than baseline for strong stimuli (reported in Figure 3B is the mean across all stimuli?).

To make things more confusing, the subsequent figures that look at the neural correlates of this behavior employ an amplitude deviant in the sensory stimuli, and not a direction deviant. While the results in Figure 3 are interesting, it is difficult to see how it naturally ties into the narrative. It may be best to instead use this task show the breadth of ways L6 CT can modulate behavior that involves deviants, as opposed to a logical next step from the gap-crossing, or even consider putting it back into the supplement and to remove the distraction altogether.

3) Beyond the first few figures, the central claim of this work is that because weak L6 CT drive changes the ensemble of active neurons in L6 and also disrupts behavior, then the original ensemble of active neurons in L6 is necessary for deviance encoding. For this kind of assertion of necessity (going beyond just sufficiency), however, requires control experiments that directly test this, often in the form of lesioning or inactivation. The authors did not perform this kind of control, which seems important for this claim. Thus, what the authors have actually proven is that the activity of the ensemble engaged after weak L6 CT drive is correlated with disrupting deviance encoding. It remains unknown whether the original ensemble is required for deviance encoding or whether the new ensemble engaged after weak L6 CT drive is rather employing mechanisms to alter deviance encoding. This difference is subtle but important in establishing the extent of claims that the data can support.

4) As described above, in addition to the trained behaviors, the employed measurement modalities also switch around from figure to figure, and even within figures in one case, making it difficult to understand what is going on. For example, Figure 4 displays a laminar silicone probe which the text describes to have been used only for the strong L6 CT drive; the text says a chronic tetrode recording was used for the weak L6CT drive results in panel B, but that is not clear. This may have come about through some practical (and non-scientific) reasons, but the authors should at least acknowledge that navigating all of these techniques is a challenge for the reader, and do what they can to help make this possible (or risk having it not read).

---

## [Author Response]

Essential revisions:While all three reviewers appreciated the general question asked, the novel approach, and the rigor of the methods, all expressed considerable concern that the conceptual presentation was far from clear. Detailed comments are listed below, but overall reviewers agreed that while further experimentation is unnecessary, better rationalization of approach, clearer connections of subtopics and methods, and explicit discussion of limitations and future directions were necessary.

We appreciate the reviewer’s thoughtful feedback and enthusiasm for the study. We followed your suggestions and made many changes in the presentation to enhance clarity. In overview:

i) We created a new Table summarizing all findings and placed this at the start of the Discussion.

ii) We apologize for any confusion about stimulation paradigms: They are far less disparate than they seemed. We now make this clear throughout the text, including:

a) All non-behavioral figures employed an amplitude deviation, applied to an ongoing 10 Hz deflection pattern, introduced between the second – seventh pulses. We are now explicit about this fact, including in the text and in images appended to almost all Figures diagramming the exact stimulation used.

b) To clarify the gap crossing with Deviation Task, we separated our description of this task into two distinct Figures, so the reader can better appreciate the effect of deviants on behavior (Figure 1), and the effect of weak optogenetic drive on deviance detection (Figure 2). Each new Figure has a parallel version of a new descriptive cartoon explicitly diagramming these.

To directly meet the reviewer’s specific suggestion, we now also present in the main text a well-controlled Motion Detection with Deviation task, in which the vibrissae of head posted mice were given a well-controlled stimulus that is modulated by inclusion of a deviance (Figure 3,). We use this task extensively in the lab (e.g. Siegle et al., 2014; presence vs. absence of vibrissal motion). In this paradigm, a directional deviant was included in the 10 Hz stimulus train. This ‘wrinkle’ of changing direction in the stimulation pattern enhances detection even though the animals are not searching for a deviant, or for a specific direction.

Figure 3: Effect of weak L6 drive on head-fixed stimulus detection with deviants. This Motion Detection with Deviation task provides a well-structured complement to the sensitivity to sensory deviations evident during the Gap Crossing with Deviation Task, which is more naturalistic but of course less controlled.

How do our stimulus conditions relate to each other?

We chose the head-fixed stimulus design in our study as simplification of the Gap Crossing with Deviation Task. Sensing that a platform is at a different position likely includes a variety of sensory parameters, predominantly changes in contact force/amplitude, and changes in contact timing, and direction. We therefore employed trains of deflections at 10Hz, with amplitude deviants in order to probe deviant encoding in a controlled yet naturalistic regime. Amplitude deviants also provide the additional benefit of avoiding stimulus-specific adaptation that would have made it harder to interpret the laminar encoding, as we now describe in the main text.

We had relegated our head-posted detection task results to the Supplementary data because we did not want to create confusion about our stimulation paradigms—this sub-study is the only place in the paper we presented directional deviants, otherwise all findings are variations on amplitude. We hope that re-inclusion of this Figure does not add confusion about paradigms—again, we tried hard to ensure this is not the case—but rather adds to the important generality in our findings. New text added to the Discussion of the detection task figure should, we believe, address this concern.

Before we list the specific critiques, we have included the general comments of each reviewer so that the authors fully appreciate the common concerns:For each experiment, the authors have chosen the appropriate recording method. Imaging deep to find sensory-responsive L6 neurons, characterize their encoding of amplitude, and then show remapping of the sensory-responsive neurons with weak optogenetic drive was particularly effective. In cases where they make arguments about encoding across the population, e.g. for deviance encoding in L2/3, they thoroughly describe response distributions with appropriate statistics and interpret them honestly in the text.The weak optogenetic stimulation used to reveal deviance encoding in L6 and L2/3 is a potential concern. It is the key manipulation in the work, and so readers must be thoroughly convinced of its meaning. The authors distinguish this manipulation from stronger drive by comparing sensory-evoked firing rates and sensory detection behavior with weak and strong optogenetic stimulation. Strong optogenetic drive blocks sensory responses across the cortical column and impairs sensory detection behavior as shown before,

As a point of clarification, we are not aware of any prior studies demonstrating suppression of sensory detection, or in our case, uninstructed sensory behavior (Gap Crossing) with high levels of L6 drive. We agree that our finding in SI sensory responses could be expected, due to the previous report of suppression of firing rates in V1 using a similar L6 activation, but the behavioral data under ‘strong’ conditions in Figure 1 are an entirely new contribution. We did not overly emphasize this particular insight because we think it mostly serves as a contrast to the more important finding here, that subtle layer 6 stimulation has selective impact for deviant stimuli, not just a full “loss of function” result such as that of the strong drive.

but weak drive does not appear to impact these measures of baseline sensory encoding (except L4 FS cells). However, authors should mention the decrease in firing rate at laser onset in L2/3 RS, L5 RS, and L4 FS cells, as it reflects an increased change in firing rate in response to the first sensory stimulus.

To directly meet this concern, we added an explicit discussion of these onset effects in the main text: We are careful to point out that while the mean sensory-evoked firing rate is not impacted by weak L6 drive, there are pre-stimulus effects and layer 4 FS response window firing rates are modulated.

We believe that the dynamics of how the cortical circuit arrives at its steady-state response to the weak L6 drive are a potentially interesting avenue for studying the associated cortical circuitry in the future: In the already quite full paper here, we did not want to dilute the main message by extensively unpacking this “baseline” effect beyond noting it to meet the reviewer’s appropriate concern, as we have now done.

While it is clever to use this manipulation to show L6's importance to deviance encoding, it is overinterpreted in the discussion. The claim that weak optogenetic drive activates a separate circuit is not supported by the data. The second claim that an ensemble of L6 neurons is used to encode deviance in L2/3 is suggested by the surprising remapping of sensory responsive L6 neurons, but it is not directly proven. These statements should be toned down to better match the findings in the paper.

To directly meet this concern, we state that while we observe L6 receptive field remapping with weak drive, and the loss of deviance-specific receptive fields in L2/3, we have not determined the specific circuit mechanisms linking these events beyond circumstantially. To quote the Discussion:

“Regardless of mechanism, our findings do not provide evidence for or against a specific circuit or circuits for change detection, but rather that specific activity patterns in L6 are required for the cortical architecture to perform change detection.”

The questions being asked are important, the experiments and study novel and really well done, and the findings exciting and likely to spawn a range of follow up questions and studies. I think that the organization of the manuscript is sub-optimal, often making it confusing, and alternately dense vs. vague. Further, there is a somewhat confusing central result – that optogenetic activation (and not inactivation) of L6 seems to disrupt deviance encoding, which in itself is not a problem (most results related to complex circuitry are confusing). But the text asserts that L6 neurons are important for deviance encoding, suggesting that activation of L6 should enhance deviance encoding in some way, which is not the finding.

We apologize for the unclear presentation of our Results in the initial submission. The reviewers are correct that low-power drive of L6 disrupts deviance coding. However, the key point of our findings is that this weak drive leads to no change in mean firing rates across L6, but instead changes which L6 cells can fire. Our findings therefore indicate that activity of a specific ensemble of L6 neurons is required for change detection, but not for sensory processing of predictable stimuli. The lesson from the strong optogenetic condition—as shown in Figure 1 and Figure 4A—is that an overall increase in L6 spiking in fact shuts down the SI column entirely and therefore generally impairs sensory-driven behavior (e.g. Gap Crossing), studying sensory deviants in conditions of such increased L6 activity is therefore made impossible by a fairly massive main effect of sensory suppression.

To directly meet this concern, we titled Figure 5:

“Figure 5 Weak depolarization of L6 CT maintains overall rates but changes the identity of stimulus-driven ensembles in L6”

In the adjoining text, we emphasize that weak activation of L6 does not change overall spiking in those cells, but rather changes which specific ensemble is recruited by sensory stimuli.

We also believe that the new table added to the Discussion helps to place all findings in context and summarize them, and hopefully will add to clarity on this point.

Most importantly, as discussed in more detail below, the central behavioral task does not seem well-suited for the questions being probed (and is essentially abandoned after the first figure – see my recommendations below). The Abstract and Introduction establish an expectation for the reader that a behavioral task targeting this direction should have a very clearly measurable role for a sensory deviant, upon which the rest of the manuscript revolves. Although the behavior used may be somehow indirectly related to this issue, it is quite vague in precisely how, and seems forced into the study.

We hope that by separating different aspects of our Gap Crossing findings into distinct Figures and providing a better description of the results—including the detailed cartoon now included in Figure 1 and Figure 2—we make clear that the discrete change in position of a platform in the middle of a bout of vibrissal sampling is a sensory deviation, implemented during natural behavior. We agree with the reviewer that this task suffers from the ambiguity of all naturalistic tasks—as with the robust and ongoing debates around natural sensing with vibrissae, the exact components of vibrissal contact that create sensation are still poorly understood: In this task, which component of vibrissal information is important--amplitude change? directional slip? timing change?

However, we strongly feel that anchoring the relevance of our data in the embodied, natural condition is an important and ethologically relevant starting point. We then aimed to re-create the most likely, and tractable aspect of this behavior, amplitude changes in the middle of a train of 10Hz deflections in our head-fixed experiments, to examine the neural coding aspects of the effect in a well-controlled setting.

At the reviewer’s suggestion, we better clarified our motivations for conducting the Gap Crossing with Deviation Task, and we also now include the head-posted task. This better-controlled condition replicates the observation that weak L6 drive impacts the behavioral benefit of deviance encoding— in this case, basic detection.

In addition, the rest of the manuscript suffers at times from switching back and forth between different measurement modalities (ephys and 2P imaging) and stimulus design (single deviant, multiple deviants in a row), making it quite challenging as a reader to digest. In the spirit of trying to help the authors sharpen their argument and make the most impact in this really important and prescient issue, the following set of comments are offered:

Our choice of 2P imaging was driven by the need to study the sparse and genetically-identifiable population of L6 CT cells that we manipulated optogenetically. We recorded over 1000 neurons using tetrode recordings, and still only found a handful of L6 receptive fields.

We now explain more explicitly and show in the Table that in all receptive field mapping studies, irrespective of measurement modality, we used an amplitude deviant that breaks the consistency of a 10 Hz train of vibrissa deflections. We now make clear that throughout all electrophysiological analyses in the manuscript, we always analyzed only the first deviant stimulus, subsequent data were discarded.

While 2P enabled a crucial kind of experiment that would otherwise have been impossible, it also required a modification to this paradigm and analysis approach: We could not distinguish responses to individual vibrissa deflections because of our low sampling rate, and as the Reviewer can appreciate, deep imaging of highly sparse ensembles with GCaMP required a more robust mapping approach.

Therefore, we chose a nearly identical stimulus design, in which we either maintained a stimulus amplitude throughout a 10Hz train of deflections, or switched from one stimulus amplitude to a second, higher or lower one. We then analyzed how these trains are encoded, without making statements about individual vibrissa deflections. We hope that the text is now clear in this point.

Again, we thank the reviewers for the time and effort to make the manuscript more straightforward to digest. We have made numerous edits to the text and believe it is now more accessible.

1) Behavior – The authors make a really nice argument for the potential role of L6 cortical neurons in deviance detection, which motivates the study very well. They then propose an optogenetic manipulation of L6 neurons, in the context of a deviance detection task in mice. However, the behavior task that was implemented in Figure 1 is certainly not the most obvious choice for a deviance detection task, and it is really puzzling why this particular task was chosen (although one may still like the task). If the task is understood correctly, the mouse is trained to whisk across a gap, to contact a platform to which they can cross the gap for a reward. The deviance is created by retracting the platform by 2mm at random times during bouts of tactile sampling. Although they authors demonstrate that this has a demonstrable effect on behavior, in that they slow their approach to cross the gap, it is really unclear how the task might be affected by the deviance, per se, as in the classical implementations of deviance detection.

We recognize that we did not sufficiently explain the reasoning for this task. To directly meet this concern, we have now broken these data into Figure 1 and Figure 2, and we now provide a detailed cartoon illustrating the findings, in addition to inclusion of the basic data and summary of these findings in a new Table.

We chose Gap Crossing because it provides a straightforward way to study deviant sensing in untrained mice: First, the behavior is uninstructed, it requires cortex (Hutson and Masterton, 1986), it is sensitive to small changes in sensory processing (Celikel and Sakmann, 2007), and findings made in this context are more likely to apply to animals in other natural contexts.

The deviant in this task is small change in platform position during the middle of a bout of whisking sampling. As in any naturally occurring tactile exploration of an object, mice are approaching the platform while they locate it, adapt their whisking pattern, and each vibrissa contact is different (Voigts et al., 2015). Mice spend more time localizing a platform to compensate if information about its position is lower fidelity (Celikel and Sakmann, 2007). Mice react to the platform position deviation used here by slowing their approach and spending more time locating it. They only react to changes in ongoing bouts of whisking, not if the platform is moved by 2mm before (or after) they palpate it (Panels B,C of the Figure and (Celikel and Sakmann, 2007; Voigts et al., 2015)).

By testing the specificity of our manipulation in this naturalistic context we therefore exclude that the relationship between weak L6 drive and behavior or electrophysiology is only specific to adaptation to unnatural static stimuli, or can only occur in absence of motor activity in headfixation, or only applies in cases where fast reaction times are required, or for the exact stimulus amplitudes we selected…etc. In sum, the task provides strong evidence that our findings generalize to natural behavior.

To perform electrophysiology and imaging experiments, we then switched to a head-fixed setting to be able to average over large number of trials with identical sensory inputs. We chose the stimulus parameters in these cases to approximate Gap Crossing. We therefore now structure our study to explain that we establish the ethological relevance of our result in naturalistic behavioral responses to deviant stimuli, and then replicate this behavioral effect in a head-fixed setting (the new Figure 3) and then in head-fixed electrophysiology and imaging experiments to best identify neural correlates.

One would have expected a more focused behavioral paradigm, where the effects of the deviance is explicitly measured through standard psychophysical means. It is true that the authors apparently tested this in such a task, as discussed in Supplemental Figure 4, but the main behavioral task of the paper presented in Figure 1 seems not well-aligned with the goals of the manuscript in a way that readers may find confusing.

At the reviewer’s suggestion, to address any concerns about how our behavioral findings relate to the head-fixed electrophysiology or imaging results, and how they extend to behavior in nonnaturalistic settings, we have now included in the main text a figure showing the same effect, specific disruption of change detection, in a simple head-fixed detection task (new Figure 3).

To be more specific, the inputs to the whisker system as the animal is actively palpating the platform (and move their body and head) are not clear, and it is not clear that the movement of the platform would create a deviant input consistent with how this is being described throughout the manuscript.

We agree that as the reviewers point out, we did not measure the exact sensory inputs in our Gap Crossing with Deviation Task, as might eventually become possible with high-speed 3D whisker tracking and associated modeling of vibrissa kinematics. We do know when vibrissae touch the platform, and we use this data to identify ‘sham-change’ trials in which the platform moves too early or too late to create a proper deviant trial. We observe a clear slowing of the approach in change trials vs. these sham-change trials (Figure 2B,C). There is therefore a clear behavioral readout of deviant detection.

We also would like to point out that this task is not a change detection task—mice are not being asked to slow down when a deviant is present to get a reward—but a spontaneous uninstructed behavior in which mice react to deviations from expected sensory input by adjusting their behavior and speed, giving us the opportunity to observe whether they perceived the deviant.

As a side note, Figure 1 was found to be quite confusing and difficult to understand what was being presented – so in any case, simplifying and clarifying was suggested. But another possibility is to remove this behavior from the manuscript, as it is quite different from the rest of the data/figures. Would there be any way to pull the behavior of Supplemental Figure 4 back into the main manuscript as the central behavior?

We agree completely, and to directly meet this suggestion we separated the Gap Crossing with Deviation Task figure into two Figure 1 and Figure 2, and we added the Supplementary Figure 4 back into the main text (Figure 3) to highlight the behavioral result that employs the head-fixed/vibrissa-held conditions in which we collected electrophysiology and imaging data.

Again, there was enthusiasm about the overall arc of the story, but concerns were raised about whether the behavior really fits.2) Overall, the manuscript is somewhat confusing – and part of this is likely due to the organization. One thing that plagues the manuscript is the use of different stimuli for almost every figure and analysis. It starts in Figure 1 with the active palpation in the gap jumping task, switches to 10 Hz train in Figure 2, and then to a 10Hz train with a single deviant in Figure 4 and Figure 5, switches to a 10Hz train with multiple deviants in sequence for the 2 photon imaging (Figure 3 and Figure 6), and finally with a mixture in Figure 7.

Again, we apologize for any confusion. We used the same stimulus design for all head-fixed experiments: A 10Hz train at a stable baseline amplitude, which can then switch to a second, higher or lower amplitude. The perceived differences across figures likely arose because we analyze the effect of the baseline amplitude (we find none) in Figure 6B. We now more clearly state this stimulus design.

It is likely the case that the 2P imaging required integrating across multiple stimulus presentations for practical reasons, necessitating this design,

That is correct, GCaMP imaging is too slow to reliably analyze individual deflections in a 10Hz train, and we now more clearly explain the reasoning behind this stimulus design. We did not interpret the L6 imaging results on a per-deflection basis.

but overall, this makes for a pretty confusing manuscript to navigate for the reader, with all of this jumping around between stimulus design and recording modality.

We believe that the new Table and many adjustments throughout will meet these concerns.

A potential concern was raised about the fact that the deviant is comprised of a sequence (and presumably the appearance of the second and third stimulus in the train might be different from the first, which is the true deviant). When is the deviant no longer a deviant, and instead the new background?

The Reviewer makes a very interesting point. Just as all detection tasks are, at a deeper level, change discrimination (comparing nothing to something), all stimuli at onset violate the prediction of sustained absence of input.

There is a wealth of literature describing differences in how stimuli are processed by the non-adapted vs. adapted sensory cortex (e.g. Webber and Stanley, 2006). Here, we were interested in small stimulus deviations that occur after an expectation has been set, such as the deviants we induced in our Gap Crossing with Deviation Task. We therefore studied deviations at positions 2 and later in the stimulus trains. Only the response to the first deviant stimulus in each train was analyzed. To directly meet this concern, we include the following Discussion:

“In this study, we employed two kinds of deviants. Variations from repeating baseline stimuli applied to the vibrissae, and ethologically relevant deviations in the position of a platform in the middle of sampling by freely behaving animals. In the latter case, the stimulus statistics (mainly vibrissa identity, angles, and curvature upon touch) change continuously as the mice approach or retreat from the target platform^31^ (Figure 2—figure supplement 1). In both cases, predictive models were at some level formed within the system, driving the change in response patterns and behavior. The similar findings across these paradigms suggests that the mechanism underlying the observed effects could be involved in more general predictive models. That said, during active sensing likely all transitions in perceptual input occur against the backdrop of a working internal model, and distinctions between anticipated and perceived input may be amplified, including the onset of a stimulus following a pause in stimulation. How the present findings relate to dynamics in more naturalistic environmental statistics is an important question for further study.”

It was thought that the authors can fix this through writing and organization, but as written, it is quite difficult to navigate. Perhaps a better explanation of the "why" for the differences in the stimulus design? Again, this is all exacerbated by the jarring transition from the behavior in Figure 1 to the rest of the paper as described above in point #1.

We appreciate that the transition from freely behaving to head-fixed experiments is a significant step, and we hope that the inclusion of the head-fixed behavior into the main text (new Figure 3), together with the multiple additional changes and explanations we added, now makes the manuscript much easier to follow.

3) The central over-arching assertion is that L6 neurons are vital for deviance encoding (and perhaps setting the stage for a predictive coding framework). However, the manipulations that are done are to either selectively engage a sub-population of L6 neurons through weak optogenetic stimulation without changing overall firing rate, or to strongly activate L6 neurons with strong optogenetic drive, both of which seem to “disrupt” deviance encoding.

This is correct, both manipulations disrupt the change encoding. However, the strong L6 drive (increasing mean L6 rates) also disrupted all other stimulus encoding, while the weaker L6 drive (changes L6 ensemble, maintains mean rates) appears to only disrupt change encoding, while having no effect on gap-crossing on static gaps, stimulus detection without deviants, or sensory-evoked firing rates. We therefore conclude that specific activity patterns in L6 are required for change detection, but not other sensory function.

To make this distinction clear, we now include a schematic Table in the Discussion. We hope that this addition makes the main conclusion of our study clearer.

Unless there was a misunderstanding of something here, it would seem that the presence of active L6 neurons disrupts deviance encoding, rather than facilitating it.

It seems like there was indeed a minor misunderstanding and we apologize for the unclear presentation of our results. Our weak L6 drive changes which L6 cells are driven by the vibrissa stimulus while maintaining mean firing rates (Figure 5). As discussed above, we now made the statements of this result clearer, and were also careful not to over-state our understanding of the intervening circuitry.

We would also like to explicitly re-state that we found no encoding of deviance itself in L6 cells. In other words, bulk firing rates in L6 do not signal the presence or absence of deviants. Instead, coding in L6 is very sparse, and optogenetically shifting specific small ensembles of L6 neurons directly leads to a loss of the behavioral effect of deviants, and a loss of deviance encoding in L2/3.

This itself is interesting, but confusion exists about statements like the last sentence of the Abstract, "These findings indicate that, despite their sparse activity, specific ensembles of stimulus-driven L6 neurons are required to form neocortical predictions, and for their behavioral benefit." The title of the manuscript captures the findings – that L6 regulates this, but the other text seems to suggest that L6 facilitates this, which seems to be the opposite of the findings. If this description somehow misconstrued the results, then the authors should find a way to clarify this text.

We show that changing which L6 ensemble is active directly leads to a disruption in change detection, but not to other sensory function. We therefore conclude that specific ensembles of active L6 cells are required for change detection. We have now attempted to explain this reasoning better in the revised Discussion and throughout the text. Our findings we believe do “indicate” this conclusion— without being needlessly semantic, in the short space of the Abstract, we wish to make sure that readers take away the main implication of the study (particularly given concerns that this implication was unclear).

4) Confusion exists about the results at the core of the study. Figure 4 says there is no net relationship between the amplitude of the stimulus deviance and the change in firing rate for L4 or L2/3 neurons (Figure 4G), but then Figure 5 dissects this, only to conclude that L4 has deviance encoding (positive correlation of deviance sign and change in firing rate), but not L2/3 (Figure 5B).

This is correct, but our description and interpretation of the results was too condensed to clearly convey the findings: There is no direct encoding of deviant presence by mean firing rates in either layer. In other words, slightly changed stimulus amplitudes did not on average lead to higher rates. This finding is we think important—it deviates (sorry for the pun…) from the canonical view, that deviants always cause “bigger” responses in neocortex due to mechanisms like stimulus specific adaptation.

As we then expand in Figure 7, deviants affect firing rates in L4 as one would expect from the prior literature showing increased sensitivity to the details of a representation following adaptation: Increased amplitude of a stimulus that is a deviant drives a detectable increase in L4 firing rates, and vice versa decreased amplitudes in deviants lead to decreased rates (Figure 7B).

We believe that the revised text, with more explicit interpretations of these findings, makes these Results clearer.

5) The confusion experienced in Figure 4 and Figure 5 continued with the optogenetic manipulation of L6 in Figure 7. Why were results not presented for L4, as opposed to just recordings in L6 and L2/3, since Figure 5 shows that the deviance encoding was emergent in L4 and not L2/3?

We concentrated on L2/3 because that is where we observed change encoding. Both L2/3 and L4 encoded current deviants, but in different ways: L4 cells increased their rates for deviants that were amplitude increases, and decreased their rates for amplitude decreases. L2/3 also showed sensitivity to the deviant’s presence, but about half the cells in L2/3 followed the opposite pattern, they reduced their rates for amplitude increases and increased them for decreases. While L4 showed the effects predicted by prior adaptation studies (stimulus precision is better encoded following adaptation), L2/3 showed encoding of the specific details of the deviance.

The same findings could have emerged if L2/3 cells were not sensitive to deviation, but rather showed specific amplitude tuning (so, when a smaller stimulus is given, it is simply the “better” stimulus for some specific cells tuned to that smaller amplitude). To test for this possibility, we maintained the deviance amplitude as constant but changed the starting baseline amplitude, and again found that L2/3 encoded deviation, not absolute amplitude. Similar emergence of change-encoding in L2/3 has been reported in visual cortex (Zmarz and Keller, 2016). L2/3 thus coded stimulus history, rather than simple deviant amplitude coding in L4.

We therefore analyzed the effect of the L6 manipulation on L2/3, and found a removal of the change encoding, both of current stimuli and stimulus history.

Is it the case that there is just no effect? Furthermore, the title of Figure 7 is "Weak L6 drive disrupts both stimulus encoding in L6 and the emergence of deviant encoding in L2/3", yet the results in Figure 7B seem to suggest that when the laser is turned on to weakly drive L6 neurons, the deviance encoding emerges (as shown by the shift of the median away from zero), rather than being disrupted. It could be the case that this was not interpreted correctly during review, but in any case, readers would also likely struggle with this. Are 7A and 7B consistent?

L6 manipulation causes the following changes in L2/3 encoding:

I) L6 stimulation removes the heterogeneous encoding of deviant stimulus amplitude in L2/3. This effect is evident in the significantly reduced width of the distribution of change coefficients and significant reduction in its spread (measured as IQR).

II) L6 stimulation causes L2/3 to act like L4, and show encoding of stimulus amplitude by a direct change in firing rates with larger deviants (more spikes) versus smaller (fewer spikes). This effect is evident in the significant rightward shift of the histogram to “positive” encoding. This is a disruption of the naturally occurring deviance coding, even though the spike counts are still driven by the stimulus.

These findings are now explicitly discussed in the Figure legend and in the text.

6) In Figure 2 there are dramatic differences in the quality of recordings in panel A versus B. Was this due to the use of tetrodes in b and silicon probe in A, and if so, is this something we should be worried about for the rest of the study that relies on the laminar recordings from the silicon probe?

Yes, the new Figure 4 (formerly Figure 2) shows laminar probe recordings for our strong laser study across layers in “A”, and “B” shows tetrode recordings. We use tetrode recordings throughout the rest of the study. We now state this clearly in the text.

7) In Figure 4, the authors show the results of small variations in stimulus amplitude in L2/3 and L4 but not L6. They later show in Figure 6 the results of L6 encoding stimulus amplitude – why did they choose to present these results in separate figures?

We separated the analyses because the L2/3 and L4 data are per-deflection analysis from tetrode arrays, and the L6 result is 2-photon GCaMP imaging and uses a slightly modified stimulus paradigm, as discussed above. We now explicitly reference the tetrode findings in the discussion of the calcium indicator data.

8) It is also not clear if the pattern of stimulus trains used in Figure 4 vs Figure 6 are the same or different. For example, in Figure 4 they use constant stim trains of one amplitude vs. another, and constant stim trains with a single deviation. In Figure 6 it looks like they test a constant stim train of one amplitude followed by a constant stim train of another amplitude. Why are these results analyzed differently?

That is correct, as stated above, the time scale of the GCaMP6s indicator we used for 2-photon imaging did not allow us to analyze individual deflections at 10Hz, but instead gave us a read out of the activity over the entire stimulus train. We therefore modified our amplitude deviant slightly to a more simplified stimulus design and analyzed the overall stimulus information encoded by L6 CT cells rather than the per-deflection code analyzed throughout the rest of the paper.

[Editors' note: further revisions were suggested prior to acceptance, as described below.]

Reviewer #1:The authors have responded to the previous critiques, and the paper is now much more readable. The new figures highlight the exact experimental design and interpretation. Overall, it is much improved.

We appreciate the constructive suggestions, they were really helpful in guiding the clarification, and glad to see the editing helped.

However, there are still some points that are not clear, and further revision could improve comprehension.1) Subsection “Weak depolarization of L6 neurons changed the identity of stimulus-driven ensembles in L6 without changing mean firing rates across layers”. I think you need to say something like we tested the effects of "weak and strong stimuli, analogous to those disrupting Gap Crossing, etc. ", on basic sensory encoding, unless you did both sets of experiments in each mouse, in which case you might want to say that explicitly.

We added a clarification to the text that now states clearly that we carried out separate sets of experiments in different mice. “We next quantified the impact of strong (disrupts Gap Crossing) and weak (selectively disrupts the impact of deviants) optogenetic L6 drive regimes on basic sensory encoding in head-fixed mice.”

2) Figure 2, while improved, is a bit confusing, as it indicates time and space in a complex way. Might it be possible to change the depiction of the moving platform to make it appear discontinuous, perhaps with multiple overlapped shadowed positions of the platform? As depicted, one has the visual impression that a platform with a recessed portion is moving left to right.

Thank you for the suggestion, we changed the depiction of the platform by making it discontinuous and adding the same arrow as we use to indicate platform motion throughout the rest of the figure.

Reviewer #2:In this work, the authors set out to address the functional role of cortical Layer 6 in predictive coding in the context of a deviant detection task in the awake, freely behaving mouse. They specifically put forth the idea that this particular circuit may be involved in predictive coding, where predictions are made based on past experience, and compared with incoming signals, perhaps driving the detection of anything that deviates from the expectation. As with the original manuscript, the general theme of linking L6 CT and predictive coding is very intriguing, and while there are increasing numbers of studies investigating the function of L6 CT neurons due to the new tools enabling the specific targeting of this cell type, the actual role of these neurons in behavior and perception is largely an area of speculation at this point. So, we really need studies like this. Moreover, the authors present very interesting results, such as a change in behavioral performance due to low L6 drive and a complete shutdown of sensory-based performance due to strong L6 drive, and the engagement of a different ensemble of neurons in L6 with weak optogenetic drive of L6 (that was also linked to weaking of behavioral sensitivity to deviant stimuli, but not affecting behavioral performance).Following the first round of critiques, the authors made substantial changes to address the concerns of the reviewers, making an earnest attempt to clarify and streamline the argument. The revisions conducted by the authors were very helpful in making the objectives and findings much more clear.

We are gratified that our objectives and findings are clearer. Finding our path to make it accessible with this body of work has been tricky, so we appreciate very *much* the time and specific perspective of this reviewer.

Perhaps the most important response for this reviewer: We failed to make a connection in the prior paper version that was obvious to us, but we now realize was never made explicit. Our Gap Crossing with Deviants task has been characterized extensively using high-speed video by our group (Voigts and Celikel, 2015), and we know that amplitude change is a major altered variable in the contact response when the platform is moved. This detailed finding from observing the vibrissae was a/the driver of our choice to look at amplitude deviation in the more controlled head posted conditions. We failed to make this quite direct connection clear before, but now do so, integrating the Gap Crossing task with the rest of the paper in a much more explicit and logical way.

However, this also highlighted some issues that are still troublesome:1) With the revision, the description of the task is now much more clear – the breaking of the original Figure 1 into the new Figure 1 and Figure 2 and the corresponding text around this improve the clarity a lot. Furthermore, the behavioral consequences of the L6 CT activation are clear – the movement of the target platform leads to a slow-down in the crossing, and weak ChR2 activation of L6CT neurons restores the behavior to the normal speed (the same as in the absence of the platform move).What I still find to be a stretch is the connection of this behavioral paradigm to the concept of deviance detection. While the behavioral results are unambiguous – as shown in Figure 2C,E,G, the nature of the sensory (whisker) stimulus during this behavior is still hazy.

We’d like to thank the reviewer for the thoughtful feedback. Applying the term ‘deviance detection’ to the Gap Crossing experiment is indeed not straightforward because in this task mice are not explicitly trained to detect deviant stimuli for a reward, but rather the detection of the deviant leads to an observable nontrained intrinsic behavior, in this case additional sampling of the platform with vibrissae before gapcrossing.

We now make sure to specify that this is *not* a trained deviant detection task, “Mice were not rewarded for detecting this deviation”, but a task in which we observe the detection or non-detection of a stimulus,

The figure shows cartoonish representations of the whisker inputs as the mouse actively contacts the platform before and after the movement of the platform, showing a sequence of identical amplitude whisker deflections, followed by a dramatic reduction in the amplitude of the whisker input after the platform moves, which could be thought of as a deviant input, but whether the whisker input actually looks like this or not, is entirely unclear. This may indeed be the case, but I think just suggesting it without showing it, leads to some discomfort about the connection.

This is a good point and we now make sure to state explicitly that none of our conclusions require a direct connection between the stimuli that lead to the deviance detection in the Gap Crossing task, and the rest of the paper.

We do, in fact, know a fair amount about what vibrissal contact dynamics in this task look like at high speed and spatio-temporal resolution, but did not emphasize this explicitly enough in the prior draft. Voigts and Celikel, 2015 (Voigts is the first author on the present study) quantified whisker dynamics in this exact task using high-speed videography. We chose this task because it is an example of mice reacting to deviant stimuli in a real-world setting.

We now make this motivation for the choice of this task explicit, and make clear the specific components of the task that change, and highlight this now.

– We state that we chose this task because it was characterized in detail previously: “.a naturalistic and untrained, but well studied and characterized sensory decision-making task”.

– Point out that contact amplitude was one of the key variables of vibrissal contact that change as a result of the sudden repositioning of the target platform: “We chose amplitude deviants because they are a significant factor in the Gap Crossing Deviants experiment^33^, provide a single parameter that can be kept close to neutral, and keeps the timing of the stimuli identical.”

– In the Figure Legend, we now specifically refer to the prior paper and the Figure within it that specifically characterizes the variable analyzed here *“*To create a sudden, small sensory deviation, in the Gap Crossing with Deviation task the target platform was retracted by ~2 mm during bouts of tactile sampling. Prior high-speed videography studies have shown this small, unexpected deviation in the position of the platform leads to changes in vibrissal motion, including reduced deflection amplitudes (Figure 9 of reference 33).”

– We also continue to make sure we emphasize that in this realistic task, we do not know the exact variables driving the behavior. “Detecting a sudden change in platform position in the Gap Crossing with Deviation experiment likely results from a combination of multiple sensory parameters that change because of the changed position of the vibrissa contacts. Previous high-speed videography studies showed that these parameters include a change in amplitude of vibrissal contact deflections (^33^ same experimental conditions), as well as other parameters such as relative timing^33^, and likely deflection angles^36,37^, assymetry^38^, timecourse of the torque^39^, and vibrations^37,40^.”

I made this comment in the first round, and while I do think the authors took my comment seriously, they seem to fundamentally disagree. So, I would just like to restate this, in the spirit of helping them understand why readers may not easily lock on to this connection. For me, the transition between the first part of the paper and the rest is still quite jarring. Overall, I love the gap-crossing paradigm, and I think there is something important happening here when the hesitation induced by the platform move is removed with L6CT activation, but I can't help but see this as disconnected from the rest of the paper.

As the reviewer points out, the deviant stimulus in the gap-crossing experiments, some change in vibrissal contact amplitude / timing / phase / direction, is not the same as in the head-posted setting. However, as we outlined above, the vibrissa-based sensory input in this setting has been characterized, and amplitude deviants are a major component. We failed to make this direct connection clear (at any level!) in the original paper, and we now do so.

Regarding our choice of head posted experiment, we chose a stimulus design that is tractable, has one degree of freedom, and aims to capture some of the properties of the gap-crossing, and arrived at relative amplitudes as the best compromise. Of course, any direct comparison between the head fixed and free experiments is going to be insufficient—for example, tension in the follicle is inarguably different in the two cases, even if we matched the external vibrissal mechanics.

We therefore avoid any conclusions that rely on direct comparisons between the two. Instead, we show that the same high-order effect, the removal of change detection, persists between these conditions. The replication of effects across very different kinds of tasks, is we think a major strength of the paper, now that the explicit link to reduced amplitudes is clearer.

We now emphasize even further that the more controlled if less natural task is not meant to somehow capture all the variables, just one prominent one.

2) Overall, this paper is a tough read. I believe this stems from the use of multiple behavioral tasks (Gap Crossing, whisker direction deviance, whisker amplitude deviance), and the jumping around between different cortical layers and measurement techniques. For example, with the behavior, related to point 1 above, Figure 3 discusses a direction deviant task, and the authors attempt to lead the reader to think this is a natural next step following the Gap Crossing task (saying that the platform move likely changes a lot of things, including amplitude and direction of motion, but none of this is shown). Even if the Gap Crossing results occur because the first (and maybe subsequent) whisker contacts following the platform move are different (smaller?) than those before the move, it is really hard to see why the direction deviance task would be the logical follow-up to this.

As outlined above, and in the paper, we agree that there are multiple aspects of vibrissal contacts that change in the platform position deviant. We also agree that the direction deviant is not as closely aligned with this task as the amplitude deviants.

We included the direction deviant task in the main text at the specific request of the reviewers in the last round of revision and did not intend to draw a direct connection between the task other than as a replication of the effect of weak L6 drive on deviant processing in another task. As mentioned above, communication is tricky in science—many different paths through a body of data could be ‘correct’ for one reader and entirely distracting for another. Given that we already put these data in at the request of this same set of reviewers, we decided to keep it in the main. More importantly, given our extensive experience presenting these data in seminars, we have found that showing these data can be a key to helping the scientific audience ‘get the point’ of this body of work (again, in agreement with the intuition of this reviewer in the last round).

To clarify this point, we have now added another specific description of this fact: “‘To test whether the impact of weak L6 drive on the effects of sensory deviation replicates in a head-fixed behavior in which one specific parameter of whisker motion is changed,…”

As a side note, I'm not really sure what to make of the fact that for the direction deviant task, the hit rate for the deviant is higher than baseline for weak stimuli and lower than baseline for strong stimuli (reported in Figure 3B is the mean across all stimuli?).

That difference is not statistically significant. In general, we do not expect the manipulation to have an effect on suprathreshold stimuli, for instance indicated by maintenance of gap-crossing performance in absence of deviants. We also specifically tested if the blue light had an effect on animal behavior by itself and found that it does not.

To make things more confusing, the subsequent figures that look at the neural correlates of this behavior employ an amplitude deviant in the sensory stimuli, and not a direction deviant. While the results in Figure 3 are interesting, it is difficult to see how it naturally ties into the narrative. It may be best to instead use this task show the breadth of ways L6 CT can modulate behavior that involves deviants, as opposed to a logical next step from the gap-crossing, or even consider putting it back into the supplement and to remove the distraction altogether.

At the request of this reviewer, we now added greater clarification that the amplitude deviants are not meant to be a direct analogue to the gap-crossing, but a quantification of the neural response to one of the major tractable parameters that change in the gap-crossing setting: “We chose amplitude deviants because they are a significant factor in the Gap Crossing Deviants experiment^33^, provide a single parameter that can be kept close to neutral, and keep the timing of the stimuli identical.”

We also explicitly state that the direction deviant task is using a different stimulus paradigm with the aim of replicating the behavioral finding (selective disruption of deviant processing) in a different, head-fixed task: “To test whether the impact of weak L6 drive on the effects of sensory deviation replicates in a head-fixed behavior in which one specific parameter of whisker motion is changed,”.

3) Beyond the first few figures, the central claim of this work is that because weak L6 CT drive changes the ensemble of active neurons in L6 and also disrupts behavior, then the original ensemble of active neurons in L6 is necessary for deviance encoding. For this kind of assertion of necessity (going beyond just sufficiency), however, requires control experiments that directly test this, often in the form of lesioning or inactivation. The authors did not perform this kind of control, which seems important for this claim. Thus, what the authors have actually proven is that the activity of the ensemble engaged after weak L6 CT drive is correlated with disrupting deviance encoding. It remains unknown whether the original ensemble is required for deviance encoding or whether the new ensemble engaged after weak L6 CT drive is rather employing mechanisms to alter deviance encoding. This difference is subtle but important in establishing the extent of claims that the data can support.

This is an interesting point, and we had not made some key terms clear enough in our manuscript. We think there are several aspects to this that we would like to clarify:

First, highly selective ensemble manipulation at the depth of layer VI is not possible, even state-of-theart 2-photon stimulation cannot reach this depth. We now state “Future studies that may achieve ensemble specific optogenetic control will be able to causally test the relationship between the encoding properties of L6 cells and their causal roles.”

Second, we would also like to point out that prior work by Scanziani and colleagues has shown that changes in overall rate of L6 activity lead to widespread inhibition or disinhibition throughout the column. We replicate the finding of inhibition via increased L6 rates in SI. These results show that increased or decreased L6 activation disrupts any basic stimulus responses.

Third, to address the question about the ensemble: We specifically do not claim that one initial ensemble, defined as a set of active cells, was required for deviance processing, in contrast to a second optogenetically driven set. What we claim is that the original state of L6, composed of an ensemble comprised of active and inactive neurons is required, and that the optogenetically driven second set (with different active and in active neurons) cannot support this function.

We make this claim based on the necessity experiments in this paper, where we modify the ensemble without affecting overall L6 rates.

We now explicitly state that: “This finding suggests that a sparse pattern of activity, comprised of both active and inactive neurons in L6, with specific connectivity, is required for deviance encoding’. and ‘In addition to the set of active neurons, the set of cells that remain inactive, or are suppressed by the vibrissa stimulus across these conditions could also play a role in the downstream decoding.”.

The suggested experiment of removing the ensemble is of course interesting, but there are technical and conceptual issues.

First, as described above, simply silencing the active cells in L6 leads to overall gain modulation and we learn nothing about the deviance-specific function of these cells.

One could then envision counteracting this effect and simultaneously inactivating the stimulus driven set of L6 cells while activating a different set if L6 cells to restore overall inhibitory drive and firing rates throughout the cortical column. This experiment is exactly the one we have carried out here, with the obvious caveat that we did not get to choose the sets of activated or inactivated L6 cells. We nevertheless found a deviance-specific disruption.

Finally, repeating the experiment with future technology that gives us a choice of suppressed and activated ensemble could give additional insight, for instance by testing if L6 cells that had certain coding properties are required but others are not etc. We now discuss this explicitly: “Future studies that may achieve ensemble specific optogenetic control will be able to causally test the relationship between the encoding properties of L6 cells and their causal roles.”

4) As described above, in addition to the trained behaviors, the employed measurement modalities also switch around from figure to figure, and even within figures in one case, making it difficult to understand what is going on. For example, Figure 4 displays a laminar silicone probe which the text describes to have been used only for the strong L6 CT drive; the text says a chronic tetrode recording was used for the weak L6CT drive results in panel B, but that is not clear. This may have come about through some practical (and non-scientific) reasons, but the authors should at least acknowledge that navigating all of these techniques is a challenge for the reader, and do what they can to help make this possible (or risk having it not read).

The choice of chronic tetrode implants (at a time before neuropixels probes were available) was motivated by the desire to sample a maximal number of neurons across different depths, and to cleanly separate regular spiking and fast spiking neurons. We added text labels to the figures that describe the recording modalities.